# The Kelch13 compartment contains highly divergent vesicle trafficking proteins in malaria parasites

**Sabine Schmidt**[ID][☯], **Jan Stephan Wichers-Misterek**[ID][☯], **Hannah Michaela Behrens**[ID], **Jakob Birnbaum**[ID], **Isabelle G. Henshall**[ID], **Jana Dröge**, **Ernst Jonscher**[ID], **Sven Flemming**[ID], **Carolina Castro-Peña**[ID], **Paolo Mesén-Ramírez**[ID], **Tobias Spielmann**[ID]*

Bernhard Nocht Institute for Tropical Medicine, Hamburg, Germany

☯ These authors contributed equally to this work.
* spielmann@bnitm.de

**Data Availability Statement:** All relevant data are within the manuscript and its Supporting information files.

**Funding:** This project has received funding from the European Research Council (ERC) under the

## Abstract

Single amino acid changes in the parasite protein Kelch13 (K13) result in reduced susceptibility of *P. falciparum* parasites to artemisinin and its derivatives (ART). Recent work indicated that K13 and other proteins co-localising with K13 (K13 compartment proteins) are involved in the endocytic uptake of host cell cytosol (HCCU) and that a reduction in HCCU results in reduced susceptibility to ART. HCCU is critical for parasite survival but is poorly understood, with the K13 compartment proteins among the few proteins so far functionally linked to this process. Here we further defined the composition of the K13 compartment by analysing more hits from a previous BioID, showing that MyoF and MCA2 as well as Kelch13 interaction candidate (KIC) 11 and 12 are found at this site. Functional analyses, tests for ART susceptibility as well as comparisons of structural similarities using Alpha-Fold2 predictions of these and previously identified proteins showed that vesicle trafficking and endocytosis domains were frequent in proteins involved in resistance or endocytosis (or both), comprising one group of K13 compartment proteins. While this strengthened the link of the K13 compartment to endocytosis, many proteins of this group showed unusual domain combinations and large parasite-specific regions, indicating a high level of taxon-specific adaptation of this process. Another group of K13 compartment proteins did not influence endocytosis or ART susceptibility and lacked detectable vesicle trafficking domains. We here identified the first protein of this group that is important for asexual blood stage development and showed that it likely is involved in invasion. Overall, this work identified novel proteins functioning in endocytosis and at the K13 compartment. Together with comparisons of structural predictions it provides a repertoire of functional domains at the K13 compartment that indicate a high level of adaption of endocytosis in malaria parasites.

## Author summary

Malaria parasites growing in red blood cells take up host cell cytosol by a poorly understood endocytic mechanism. Kelch13, the protein mutated in parasites with lowered

European Union's Horizon 2020 research and innovation programme (grant agreement No. 101021493). This grant was awarded to TS. HMB acknowledges funding by the Ortrud Mührer Fellowship of the Vereinigung der Freunde des Tropeninstituts Hamburg e.V, https://www.bnitm.de/alumni-und_freunde/vdf. CCP thanks the DAAD (https://www.daad.de/de/) for funding (Personenkennziffer 91726017). The funders had no role in study design, data collection and analysis, decision to publish, or preparation of the manuscript.

**Competing interests:** The authors have declared that no competing interests exist.

susceptibility to Artemisinin-drugs (ART), is involved in this process and the rate of endocytosis was linked to ART susceptibility. To better understand the endocytosis process we here screened further candidates identified in previous searches for proteins in proximity to Kelch13 (the K13 compartment) and found 4 additional K13 compartment proteins. Two of these had a function in endocytosis but not in early stage parasites, the stage associated with lowered ART susceptibility. One of these proteins was a Myosin, indicating involvement of actin in endocytosis in later stage parasites. Analysing the AlphaFold structures of the here and previously confirmed K13 compartment proteins revealed many with vesicle trafficking domains in unusual configurations. This indicates that the K13 compartment is involved in a highly parasite-specific vesicle trafficking process, suggesting an unusual endocytosis mechanism. We also found one K13 compartment protein with a likely function in invasion, indicating K13 compartment proteins can have functions outside endocytosis, although for this particular protein this function may also be attributable to an additional cellular location we observed.

## Introduction

Malaria, caused by protozoan parasites of the genus *Plasmodium*, is one of the deadliest infectious diseases, responsible for an estimated 627 000 deaths and 241 million cases in 2020 [1]. Malaria deaths have been declining in the last two decades [1,2], but this trend has reversed in the last few years [1]. One of the factors that contributed to the reduction of cases is efficient treatment which currently relies on artemisinin and its derivatives (ART) administered together with a partner drug [3]. However, parasites with a reduced susceptibility to ART have become widespread in South East Asia [4–12] and have now also emerged in Africa [13–16], Papua New Guinea [17] and South America [18]. The reduced effectivity of ART is linked to a delayed parasite clearance in patients that is due to a decreased susceptibility of ring-stage parasites to ART [19,20], leading to recrudescence and treatment failure in patients [21–24]. Experimental human infections indicate that a doubled clearance half-life in parasites with reduced ART susceptibility results in a >10 fold increase in viable parasite half-life compared to the sensitive parasites [25]. Decreased susceptibility of ring-stage parasites can be measured *in vitro* using the ring stage survival assay (RSA) [26]. We here refer to this *in vitro* measurable phenomenon as "*in vitro* ART resistance" when the survival of parasites is above 1% in an RSA as previously defined [26]. Point mutations in the gene encoding the parasite protein Kelch13 (K13) are the main cause for the reduced ART susceptibility of laboratory and field parasite isolates [27,28] and was shown to lead to decreased K13 protein levels [29–33]. This results in reduced hemoglobin uptake via endocytosis, which is assumed to cause less ART activation through hemoglobin degradation products, resulting in the reduced parasite ART susceptibility [29,34]. Beside K13 mutations in other genes, such as Coronin [35] UBP1 [29,36–38] or AP2μ [38,39] have also been linked with reduced ART susceptibility. In contrast to K13 which is only needed for endocytosis in ring stages (the stage relevant for *in vitro* ART resistance), some of these proteins (AP2μ and UBP1) are also needed for endocytosis in later stage parasites [29]. At least in the case of UBP1, this is associated with a higher fitness cost but lower resistance compared to K13 mutations [34,40]. Hence, the stage-specificity of endocytosis functions is relevant for *in vitro* ART resistance: proteins influencing endocytosis in trophozoites are expected to have a high fitness cost whereas proteins not needed for endocytosis in rings would not be expected to influence resistance.

Endocytosis is a critical process for blood stage growth of malaria parasites. During its asexual development in erythrocytes the parasite uses this process to take up more than two-thirds of the soluble host cell content, which consists almost exclusively of hemoglobin [41–43]. The endocytosed host cell cytosol is transported to the acidic lysosome-like food vacuole (FV), where digestion of hemoglobin results in generation of free heme, toxic to the parasite, which is further converted into nontoxic hemozoin [44,45]. Host cell cytosol uptake (HCCU) provides both space and building blocks (amino acids from digested hemoglobin) for parasite growth [42]. As a consequence, ART resistant parasites are hypersensitive to amino acid restriction, highlighting the importance of endocytosis for parasite growth and its connection to ART resistance [46,47]. Hemoglobin trafficking to the parasite food vacuole is believed to be initiated at membrane invaginations called cytostomes [30,42,48,49] followed by vesicular transport from the parasite plasma membrane (PPM) to the food vacuole [42,50], likely in an actin-myosin motor dependent manner [48,51–53]. The molecular effectors involved in this process remain poorly characterized. So far VPS45 [54], Rbsn5 [55], Rab5b [55], the phosphoinositide-binding protein PX1 [56], the host enzyme peroxiredoxin 6 [57] and K13 and some of its compartment proteins (Eps15, AP2μ, KIC7, UBP1) [29] have been reported to act at different steps in the endocytic uptake pathway of hemoglobin. While inactivation of VPS45, Rbsn5, Rab5b, PX1 or actin resulted in an accumulation of hemoglobin filled vesicles [53–56], indicative of a block during endosomal transport (late steps in endocytosis), no such vesicles were observed upon inactivation of K13 and its compartment proteins [29], suggesting a role of these proteins during initiation of endocytosis (early steps in endocytosis).

In a previous work we used a quantitative BioID approach to identify proteins in close proximity to K13 and its interactor Eps15 [29]. We designated these proteins as K13 interaction candidates (KICs) [29]. As the proteins identified by proximity labelling approaches not only include interactors but also proteins that are located in a close spatial position of the bait [58,59], we will here refer to the collective of these proteins as the "proxiome". We reasoned that due to the high number of proteins that turned out to belong to the K13 compartment when validating the top hits of the K13 [29], the remaining hits of these experiments might contain further proteins belonging to the K13 compartment. Here we identified and functionally analysed further K13 compartment proteins from the K13 proxiome [29] and classify these and previously confirmed K13 compartment proteins. Location and functional data as well as comparisons of alpha fold predicted structural elements revealed that the K13 compartment contains proteins suggestive of a highly divergent endocytosis mechanism in malaria parasites but also proteins that are dispensable for blood stage growth and at least one with an important function in a different process.

## Results

To identify novel K13 compartment proteins we further exploited previously described proximity-dependent biotinylation experiments that had used K13 or the K13 compartment protein Eps15 as bait, selecting enriched proteins not characterized in our previous work [29] (S1 Table). In order to do this we excluded proteins that (i) had previously been analysed, (ii) were either linked with or had been shown to localise to the inner membrane complex (IMC) or the basal complex (PF3D7_1345600 [60]; PF3D7_0109000 (PhIL1) [61–63]; PF3D7_0717600 (IMC32) [64]; PF3D7_0822900 (PIC2) [65], PF3D7_1018200 (PPP8) [65–67], (iii) are considered typical DiQ-BioID 'contaminants' (PF3D7_0708400 (HSP90) and PF3D7_1247400 (FKBP35) [29,68]), (iv) localised to the apical polar ring in *P. berghei* (PF3D7_1141300 (APR1) [69]), (v) localised to the nucleus PF3D7_1247400 (FKBP35) [70,71], (vi) were linked with the apicoplast (PF3D7_0721100 [72]) or (vi) were also present in BioID experiments using

Clathrin heavy chain (CHC) as bait [29] (PF3D7_0408100). These selection criteria resulted in a candidate list of thirteen proteins (PF3D7_1438400 (MCA2), PF3D7_1243400, PF3D7_1365800, PF3D7_1447800, PF3D7_1142100, PF3D7_0103100 (VPS51), PF3D7_1329100 (MyoF), PF3D7_1329500, PF3D7_0405700 (UIS14), PF3D7_0907200, PF3D7_0204300, PF3D7_1117900 and PF3D7_1016200), of which ten were chosen for further characterization in this manuscript (Tables 1 and S1).

## MyosinF is involved in host cell cytosol uptake and associated with the K13 compartment

The presence of MyosinF (MyoF; PF3D7_1329100, previously also MyoC), in the K13 proxiome could indicate an involvement of actin/myosin in endocytosis in malaria parasites. We therefore analysed its localisation in the cell. For this, we first generated transgenic parasites expressing a C-terminally 2xFKBP-GFP-2xFKBP tagged version of MyoF expressed from its original locus using selection-linked integration (SLI) [73] (S1A Fig). Expression and localisation of the fusion protein was analysed by fluorescent microscopy. The tagged MyoF was detectable as foci close to the food vacuole from the stage parasites turned from late rings to young trophozoite stage onwards, while in schizonts multiple MyoF foci were visible (Figs 1A and S2A). This expression pattern is in agreement with its transcriptional profile [74,75]. A similar localisation was observed in a cell line expressing MyoF with a smaller tag (MyoF-2xFKBP-GFP) from its endogenous locus (S1B and S2B Figs). The proximity of the MyoF foci to the food vacuole in trophozoites was also evident by co-localisation with an episomally expressed food vacuole marker (P40PXmCherry) [54,76] (Fig 1B). Next, we episomally co-expressed mCherry-K13 in the MyoF-2xFKBP-GFP-2xFKBP[endo] parasites to assess whether MyoF is found at the K13 compartment (Figs 1C and S2A). Quantification from live fluorescence images of late ring and trophozoite stage parasites revealed that 8% of MyoF foci overlapped with K13 foci, 12% showed a partial overlap, 36% were close together (touching but not overlapping), while 44% of MyoF foci were further away from K13 foci (Fig 1D). As the GFP-tagging of MyoF appeared to have some effect on the parasite (see below), we validated the MyoF localisation in respect to the K13 compartment by generating parasites with an

**Table 1. Selection of putative K13 compartment proteins for further characterization in this manuscript.**

| Gene ID | PlasmoDB annotation | Gene name |
|---|---|---|
| PF3D7_1438400 | metacaspase-like protein | *mca2* |
| PF3D7_1243400 | conserved Plasmodium protein, unknown function | |
| PF3D7_0204300 | conserved Plasmodium protein, unknown function | |
| PF3D7_1365800 | conserved Plasmodium protein, unknown function | |
| PF3D7_1447800 | calponin homology domain-containing protein, putative | |
| PF3D7_1142100 | conserved Plasmodium protein, unknown function | *kic11* |
| PF3D7_0103100 | vacuolar protein sorting-associated protein 51, putative | *vps51* |
| PF3D7_1117900 | conserved Plasmodium protein, unknown function | |
| PF3D7_1016200 | Rab3 GTPase-activating protein non-catalytic subunit, putative | |
| PF3D7_1329100 | myosin F | *myof* |
| PF3D7_1329500 | conserved Plasmodium protein, unknown function | *kic12* |
| PF3D7_0405700 | lysine decarboxylase, putative | *uis14* |
| PF3D7_0907200 | GTPase-activating protein, putative | |

Hits in cells labelled with fainter colour are from the less stringent filtering group (significant with FDR<1% in 2 out of 4 reactions of any bait [29])

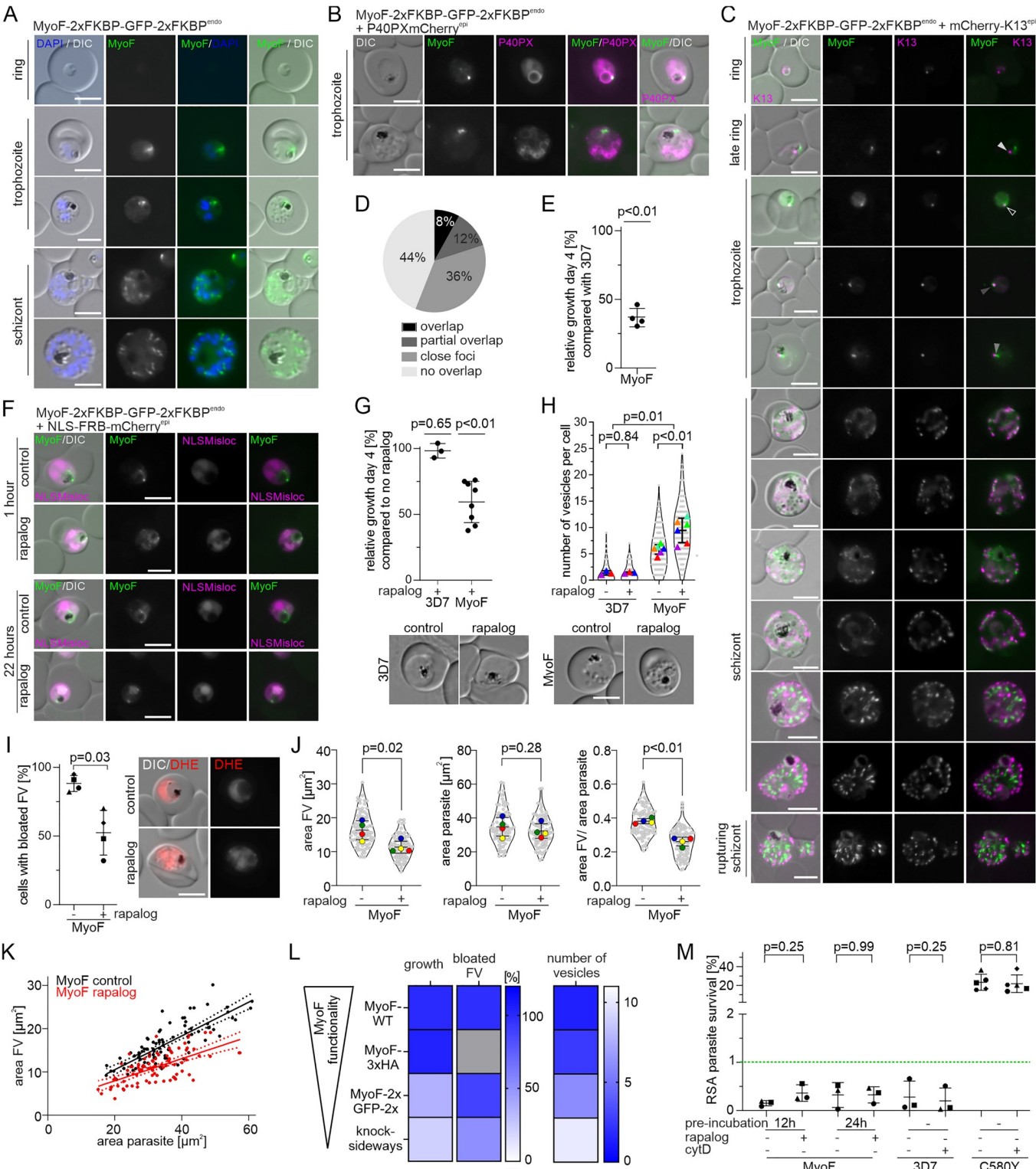

**Fig 1. MyoF is involved in host cell cytosol uptake and associated with the K13 compartment.** (**A**) Localisation of MyoF-2xFKBP-GFP-2xFKBP expressed from the endogenous locus by live-cell microscopy across the intra-erythrocytic development cycle. Nuclei were stained with DAPI. Scale bar, 5 μm. (**B**) Live cell microscopy images of parasites expressing the MyoF-2xFKBP-GFP-2xFKBP fusion protein with an episomally expressed FV marker P40PX-mCherry. Scale bar, 5 μm. (**C**) Live cell microscopy images of parasites expressing the MyoF-2xFKBP-GFP-2xFKBP fusion protein with episomally expressed mCherry-K13. Scale bar, 5 μm. Arrows indicate categories from Fig 1D. (**D**) Foci were categorized into 'overlap' (black), 'partial overlap' (dark grey), 'close foci' (= less than one focus

radius apart) (grey) and 'non overlap' (light grey). Three independent live microscopy sessions with each n = 14 analysed parasites. (E) Relative growth of synchronised MyoF-2xFKBP-GFP-2xFKBP[endo] compared with 3D7 wild type parasites after two growth cycles. Each dot shows one of four independent growth experiments. P-values determined by one-sample t-test. (F) Live-cell microscopy of knock sideways (rapalog) and control MyoF-2xFKBP-GFP-2xFKBP[endo]+1xNLSmislocaliser parasites 1 hour or 22 hours after the induction by addition of rapalog. Scale bar, 5 μm. (G) Relative growth of asynchronous 3D7 and asynchronous MyoF-2xFKBP-GFP-2xFKBP[endo]+1xNLSmislocaliser parasites (+ rapalog) compared with the corresponding control parasites (without rapalog) after five days. Three (3D7) and eight (MyoF-2xFKBP-GFP-2xFKBP[endo]) independent experiments (individual experiments shown in S2D Fig). Error bars, mean ± SD. P-values determined by Welch's t-test. (H) Number of vesicles per parasite in trophozoites determined by live-cell fluorescence microscopy (DIC) in 3D7 and MyoF-2xFKBP-GFP-2xFKBP[endo]+1xNLSmislocaliser parasites with and without rapalog addition. Three (3D7) and six (MyoF-2xFKBP-GFP-2xFKBP[endo]+1xNLSmislocaliser) independent experiments with each time n = 16–25 (mean 20.9) parasites analysed per condition. Mean of each independent experiment indicated by coloured triangle, individual datapoints by grey dots. Data presented according to SuperPlot guidelines [147]; error bars represent mean ± SD. P-value for ± rapalog determined by paired t-test and for 3D7 vs MyoF by Mann-Whitney. Representative DIC images are displayed below. (I) Bloated food vacuole assay with MyoF-2xFKBP-GFP-2xFKBP[endo] parasites 8 hours after inactivation of MyoF (+ rapalog) compared with controls (- rapalog). Cells were categorized as with 'bloated FV' or 'non-bloated FV' and percentage of cells with bloated FV is displayed; n = 4 independent experiments with each n = 33–40 (mean 34.6) parasites analysed per condition. P-values determined by Welch's t-test. Representative DIC and fluorescence microscopy images shown on the right. Parasite cytoplasm was visualized with DHE. Experimental setup shown in S2G Fig. (J) Area of the FV, area of the parasite and area of FV divided by area of the corresponding parasite of MyoF-2xFKBP-GFP-2xFKBP[endo]+1xNLSmislocaliser parasites analysed in Fig 1I. Mean of each independent experiment indicated by coloured dots, individual data points by grey dots. Data presented according to SuperPlot guidelines [147]; error bars represent mean ± SD. P-value determined by paired t-test. Representative DIC and fluorescence microscopy images are shown in the S2J Fig. (K) Area of FV of individual cells plotted versus the area of the corresponding parasite in MyoF-2xFKBP-GFP-2xFKBP[endo]+1xNLSmislocaliser parasites of the experiments shown in Fig 1H-I. Line represents linear regression with error indicated by dashed line. (L) Summary heatmap showing the effect of C-terminal tagging with 3xHA or 2xFKBP-GFP-2xFKBP and knock-sideways on MyoF functionality depicted as growth [%], bloated FV [%] and number of induced vesicles. (M) Parasite survival rate (% survival compared to control without DHA) 66 h after 6 h DHA (700 nM) treatment in standard RSA. MyoF-2xFKBP-GFP-2xFKBP[endo] parasites + 1xNLSmislocaliser were pretreated with rapalog either 12 or 24 hours prior to the assay. Three independent experiments, P-value determined by Wilcoxon test. Green dashed line indicates 1% ART resistance cut-off [26]. 2988–8392 (mean 5102) cells for control and 22704–44038 (mean 32077) cells for DHA treated samples were counted. Experimental setup shown in S2K Fig. 3D7 and 3D7-K13[C580Y] [29] parasites were incubated with CytochalasinD for 6h in parallel with DHA pulse. 850–13180 (mean 4677) cells for control and 1065–38515 (mean 15532) cells for DHA treated samples were counted. Three (3D7) or five (3D7-K13[C580Y]) independent experiments, P-value determined by Wilcoxon test.

endogenously 3xHA-tagged MyoF (MyoF-3xHA[endo] parasites) using SLI (S1C Fig) and episomally co-expressed mCherry-K13. IFA confirmed the focal localisation of MyoF and its spatial association with mCherry-K13 foci with a tendency for a higher overlap with K13, which might be due to the partial inactivation of the GFP-tagged MyoF (S2C Fig). We also detected MyoF signal in ring stage parasites (S2C Fig). As we did not detect the GFP-tagged version in rings, this likely indicates a low level of expression of MyoF in ring stage parasites as IFA is more sensitive. In late schizonts and merozoite the MyoF-GFP signal was not associated with K13, but was present in elongated foci (Figs 1C and S2A) reminiscent of the MyoE signal previously reported in *P. berghei* schizonts [77]. Taken together, these results show that MyoF is in foci that are frequently close or overlapping with K13, indicating that MyoF is found in a regular close spatial association with the K13 compartment and at times overlaps with that compartment.

During routine *in vitro* culturing we noticed that MyoF-2xFKBP-GFP-2xFKBP[endo] parasites grew poorly and subsequent flow cytometry-based proliferation assays revealed a mean relative growth of 36.7% compared to 3D7 wild type parasites after two replication cycles (Fig 1E). These results indicated that C-terminal 2xFKBP-GFP-2xFKBP tagging of MyoF impaired its function and that this protein has an important role for the growth of asexual blood stages. We therefore generated MyoF-2xFKBP-GFP-2xFKBP[endo] parasites episomally expressing a nuclear mislocaliser (1xNLS-FRB-mCherry) in order to conditionally inactivate it using knock-sideways. This system allows the conditional mislocalisation of a protein from its site of action into the nucleus upon addition of rapalog [73,78,79]. Assessment of mislocalisation efficacy by fluorescent microscopy at time points between 1 hour and 22 hours post induction revealed partial mislocalisation of MyoF-2xFKBP-GFP-2xFKBP to the nucleus with some MyoF remaining at foci close to the food vacuole (Figs 1F and S2D). Despite the only partial inactivation of MyoF by knock sideways, flow cytometry-based proliferation assays revealed a 40.5% reduced growth after two replication cycles upon addition of rapalog compared to

control parasites without rapalog, while such an effect was not observed for 3D7 wild type parasites upon addition of rapalog (Figs 1G and S2E). Hence, conditional inactivation of MyoF further reduced growth despite the fact that the tag on MyoF already led to a substantial growth defect, indicating an important role for MyoF during asexual blood stage development.

Inspection of the MyoF-2xFKBP-GFP-2xFKBP[endo], compared to 3D7 wild type-trophozoites, revealed an increased number of vesicles in the parasite cytoplasm (Fig 1H), resembling the phenotype observed after inactivation of VPS45 or Rbsn5, two proteins involved in the transport of host cell cytosol to the parasite's food vacuole [54,55]. This was even more pronounced upon conditional inactivation of MyoF by knock-sideways (Fig 1H), suggesting this is due to a reduced function of MyoF. Of note, C-terminal tagging with the small 3xHA epitope had no significant effect on parasite proliferation (S2F Fig), and accordingly showed only a small, but still significant increase of the number of vesicles in the parasite cytoplasm (S2G Fig), indicating only a minor impairment by C-terminal tagging with the small 3xHA epitope. These findings indicated that inactivation of MyoF might impair the transport of host cell cytosol to the food vacuole.

We therefore directly tested for an effect of MyoF inactivation on HCCU using a 'bloated food vacuole assay' [54]. For this we incubated parasites upon MyoF inactivation (and control parasites without rapalog) with the protease inhibitor E64 [80]. In the presence of E64, newly internalized hemoglobin cannot be degraded and accumulates in the food vacuole, resulting in bloated food vacuoles if endocytosis is operational. In 3D7 wild type parasites 98% of the cells showed a bloated food vacuole in both rapalog treated and control parasites S2H and S2J Fig). In contrast, 88% of the MyoF-2xFKBP-GFP-2xFKBP[endo] control (without rapalog) parasites developed bloated food vacuoles, and after further inactivation of MyoF by knock sideways (with rapalog) only 52% of the cells showed a bloated food vacuole, indicating reduced HCCU with increasing inactivation of MyoF (Figs 1I, S2H and S2J). As the effect of MyoF inactivation was only partial, we measured the parasite and food vacuole size to obtain a finer readout for HCCU. This analysis revealed a significantly reduced food vacuole size in the parasites with MyoF inactivated by knock sideways, while there was no significant effect on parasite size (Fig 1J). Plotting the values of the individual parasites showed that the food vacuoles of similarly sized parasites were consistently smaller in the MyoF knock sideways compared to the parasites without knock sideways inactivation of MyoF (Fig 1K), indicating that the reduced hemoglobin delivery to the food vacuole upon inactivation of MyoF was not an indirect effect due to parasite growth impairment during the assay time. Next, we plotted the endocytosis phenotypes (vesicles and bloating) and growth of 3D7 (wild type MyoF activity), MyoF-3xHA (small degree of MyoF inactivation), MyoF-2xFKBP-GFP-2xFKBP (more MyoF inactivation) and MyoF-2xFKBP-GFP-2xFKBP knock sideways (even more MyoF inactivation) which showed that the degree of the endocytosis phenotypes correlated with growth and hence level of MyoF inactivation (Fig 1L). Overall, this indicates that MyoF function is needed for hemoglobin to reach the food vacuole.

Finally, we also tested if inactivation of MyoF has an effect on *in vitro* ART resistance. RSAs [26] revealed no significant increase in parasite survival upon MyoF inactivation with neither 12 h nor 24 h rapalog pre-incubation to inactivate MyoF by knock sideways (Figs 1M and S2K). Of note, due to the partial inactivation due to the tagging of MyoF we would have expected the control (MyoF-2xFKBP-GFP-2xFKBP parasites without rapalog) to already have elevated RSA survival levels if MyoF impairment reduces endocytosis in rings, which was not the case (Fig 1M). Similarly, also incubation with the actin destabilising agent Cytochalasin D [81], had no effect on RSA survival in 3D7 or K13[C580Y] [29] parasites, indicating an actin/myosin independent endocytosis pathway in ring stage parasites (Fig 1M) and also speaking against a function of other myosins in endocytosis in rings.

Overall, our results indicate a close association of MyoF foci with the K13 compartment and a role of MyoF in endocytosis albeit not in rings and at a step in the endocytosis pathway when hemoglobin-filled vesicles had already formed and hence is subsequent to the function of the other so far known K13 compartment proteins.

## KIC11 is a K13 compartment protein important for asexual parasite proliferation due to a function not involving endocytosis

PF3D7_1142100, currently annotated as 'conserved Plasmodium protein, unknown function', was renamed K13 interacting candidate 11 (KIC11), following the previously established nomenclature for potential K13 compartment proteins [29]. In order to test whether KIC11 is a member of the K13 compartment or not, we first generated transgenic parasites expressing it as a C-terminally 2xFKBP-GFP-2xFKBP tagged fusion protein from the original genomic locus using SLI (KIC11-2xFKBP-GFP-2xFKBP^endo parasites). Correct genomic modification of the *kic11* locus was verified by PCR (S1D Fig). The tagged KIC11 showed multiple foci in ring and trophozoite stage parasites (mean 2.2 per ring stage and 4.0 per trophozoite), while in schizonts many KIC11-2xFKBP-GFP-2xFKBP foci were visible (mean 14.3), which are not co-localizing with apical organelle marker proteins (Figs 2A, S3A and S3B). Next, we generated KIC11-2xFKBP-GFP-2xFKBP^endo parasites episomally co-expressing mCherry-K13 to assess presence at the K13 compartment. Fluorescence microscopy revealed more KIC11 than K13 foci per cell in ring and trophozoite stage parasites (S3B Fig). Quantification showed that in ring stage parasites 16% of K13 foci overlapped with KIC11 foci, 45% showed a partial overlap, 26% were close together (touching but not overlapping), while 13% were further away. In trophozoites 23% of K13 foci overlapped with KIC11 foci, 46% showed a partial overlap, 15% were close together, while 15% were further away (Figs 2B, 2C and S3C). We conclude that KIC11 is a protein located at the K13 compartment with an additional protein pool located at foci elsewhere.

In order to assess the importance of KIC11 for parasite proliferation, we generated KIC11-2xFKBP-GFP-2xFKBP^endo parasites episomally co-expressing the nuclear mislocaliser 1xNLS-FRB-mCherry, enabling conditional inactivation by knock-sideways. Addition of rapalog resulted in efficient mislocalisation of KIC11-2xFKBP-GFP-2xFKBP into the nucleus as evaluated 4 and 16 hours post induction (Fig 2D). Assessment of parasite proliferation by flow-cytometry over two developmental cycles revealed a mean relative growth of 10.3% compared to control parasites, indicating an important function of KIC11 for asexual parasite proliferation (Figs 2E and S3D). This interpretation was supported by several unsuccessful attempts to generate a cell line with a truncated *kic11* using the SLI-TGD system [73].

To better characterize the growth phenotype of the KIC11 knock-sideways, we added rapalog to tightly synchronised parasites at different time points (4, 24, and 32 hpi) and monitored parasite growth by flow cytometry. Additionally, we quantified parasite stage in Giemsa smears at 6, 24, 32, 40, 48, 72, and 96 hpi. While no effect on parasitemia and stage distribution was observed during growth in the first cycle, a reduced number of newly formed ring stage parasites was obvious at the beginning of the second cycle, indicating an invasion defect or an effect on parasite viability in merozoites or early rings but no effect on other parasite stages (Fig 2F, 2H, S3F and S3G). As KIC11 appeared to be important at the schizont to ring transition, we further analysed the phenotype at this stage. KIC11 inactivation did not affect egress but resulted in fewer detectable rings, suggesting a defect in invasion or possibly very early ring stage development (Figs 2H and S3G). No consistent overlap of KIC11 foci with rhoptry (ARO) or microneme (AMA1) marker proteins was observed, while as expected [65] for a K13 compartment protein, an overlap with the IMC marker (IMC1c) was apparent. In addition,

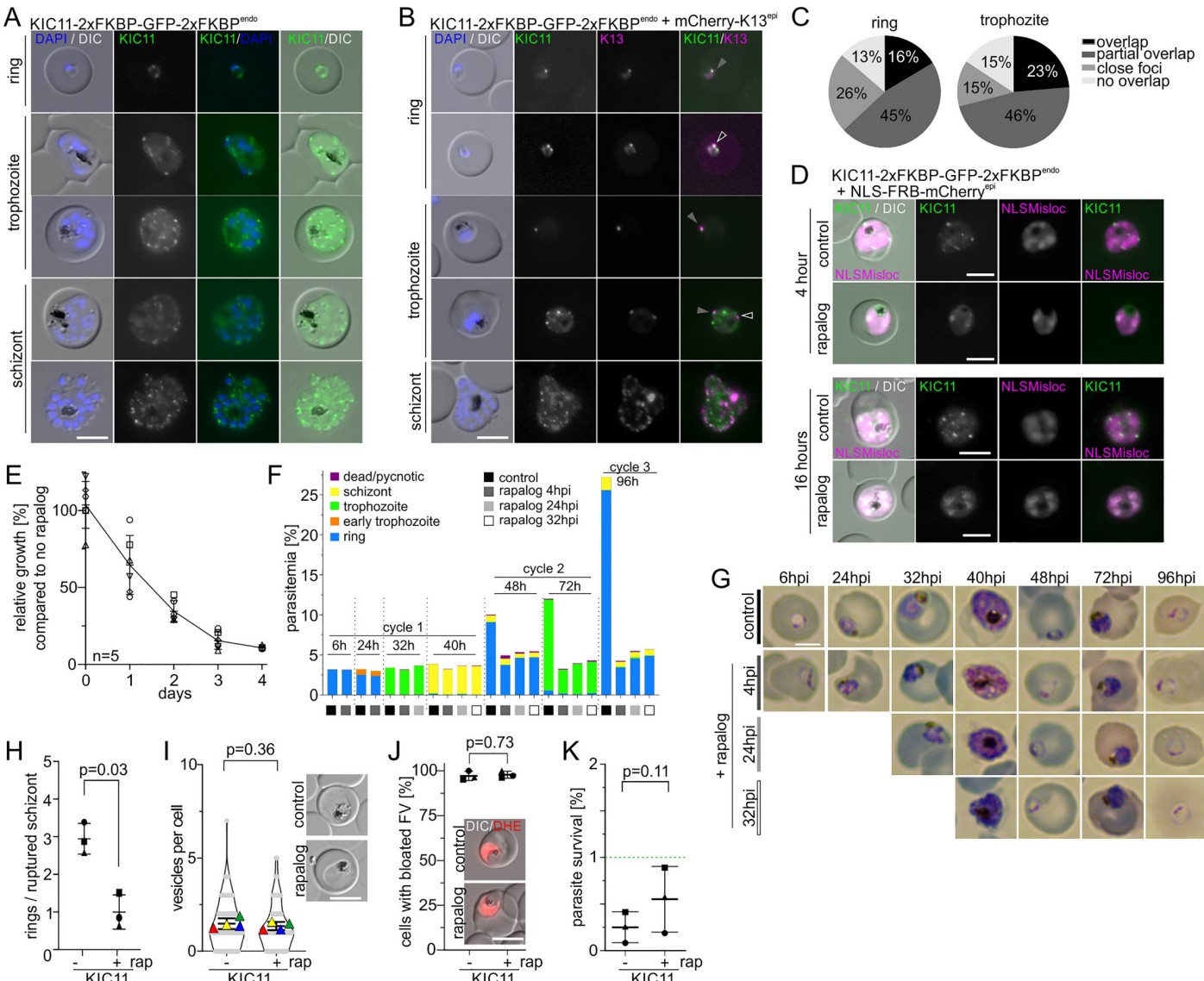

**Fig 2. KIC11 is a K13 compartment protein important for asexual parasite proliferation, but not involved in endocytosis or *in vitro* ART resistance. (A)** Localisation of KIC11-2xFKBP-GFP-2xFKBP expressed from the endogenous locus by live-cell microscopy across the intra-erythrocytic development cycle. Nuclei were stained with DAPI. Scale bar, 5 μm. **(B)** Live cell microscopy images of parasites expressing KIC11-2xFKBP-GFP-2xFKBP with episomally expressed mCherry-K13. Nuclei were stained with DAPI. Scale bar, 5 μm. Arrows indicate categories from Fig 2C. **(C)** Foci were categorized into 'overlap' (black), 'partial overlap' (dark grey), 'close foci' (= less than one focus radius apart) (grey) and 'non overlap' (light grey) for ring (n = 35) and trophozoite (n = 25) stage parasites. **(D)** Live-cell microscopy of knock sideways (+ rapalog) and control (without rapalog) KIC11-2xFKBP-GFP-2xFKBP<sup>endo</sup>+1xNLSmislocaliser parasites 4 and 16 hours after the induction of knock-sideways by addition of rapalog. Scale bar, 5 μm. **(E)** Relative growth of asynchronous KIC11-2xFKBP-GFP-2xFKBP<sup>endo</sup>+1xNLSmislocaliser plus rapalog compared with control parasites over five days. Five independent experiments were performed (depicted by different symbols) and mean relative parasitemia ± SD is shown (individual experiments shown in S4A Fig). **(F)** Parasite stage distribution in Giemsa smears at the time points (average hours post invasion, h) indicated above each bar in tightly synchronised (±4h) KIC11-2xFKBP-GFP-2xFKBP<sup>endo</sup>+1xNLSmislocaliser parasites (rapalog addition at 4 hpi, 20 hpi, or 32 hpi and control) assayed over two consecutive cycles (last time point in cycle 3). A second replicate is shown in S3F Fig. **(G)** Giemsa smears of control and at 4 hpi, 20 hpi, or 32 hpi rapalog-treated KIC11-2xFKBP-GFP-2xFKBP<sup>endo</sup>+1xNLSmislocaliser parasites shown in (E). **(H)** Quantification of rings per ruptured schizont at 'post-egress' time point compared to 'pre-egress' time point in knock-sideways (+rapalog (rap)) compared to control (-rapalog) KIC11-2xFKBP-GFP-2xFKBP<sup>endo</sup>+1xNLSmislocaliser parasites from n = 3 independent experiments (indicated by different symbol shapes). P values were determined with ratio-paired t-test, data displayed as mean ±SD. Quantification of corresponding ruptured schizonts is shown in S3G Fig. **(I)** Number of vesicles per parasite in trophozoites determined by live-cell fluorescence microscopy (DIC) in KIC11-2xFKBP-GFP-2xFKBP<sup>endo</sup>+1xNLSmislocaliser parasites with and without rapalog. Four independent experiments with n = 16–59 (mean 30.1) parasites analysed per condition per experiment. Rap, rapalog. Mean of each independent experiment indicated by coloured triangle, individual data points by grey dots. Data presented according to SuperPlot guidelines [147]; error bars represent mean ± SD. P-value determined by paired t-test. Representative DIC images are displayed. **(J)** Bloated food vacuole assay with KIC11-2xFKBP-GFP-2xFKBP<sup>endo</sup> parasites 8 hours after inactivation of KIC11 (+rapalog (rap)) compared with control (without rapalog). Cells were categorized as with 'bloated FV' or 'non-bloated FV' and displayed as percentage of cells with bloated FV; n = 3 independent experiments with each n = 19–36 (mean 26.5) parasites analysed per condition. P-values determined by Welch's t-test. Representative

DIC and fluorescence microscopy images are shown in the right panel. Parasite cytoplasm was visualized with DHE. Experimental setup shown in S3H Fig. **(K)** Parasite survival rate (% survival compared to control without DHA) 66 h after 6 h DHA treatment in standard RSA. Three independent experiments, P-value determined by paired t-test. Green dashed line indicates 1% ART resistance cut-off [26]. 2896–7135 (mean 4502) cells for control and 23183–32455 (mean 28496) cells for DHA treated samples were counted. Rap, rapalog. Experimental setup shown in S3I Fig.

KIC11 inactivation did not result in a detectable alteration of these markers, excluding major structural disorganisation of the micronemes, rhoptries or IMC as a reason for the observed phenotype (S3A Fig).

As trophozoites with inactivated KIC11 developed into schizonts without morphological differences to controls (Figs 2G and S3A), KIC11 does not appear to be needed for endocytosis. To directly address this we tested if inactivation of KIC11 influences *in vitro* ART resistance (based on RSA) or endocytosis (using vesicle accumulation and bloated food vacuole assays), but no significant differences were observed (Fig 2I and 2K).

Overall, our results indicate that KIC11 is part of the K13 compartment but also in additional foci and that it has an important role in invasion or very early ring stage development, which is in contrast to previously characterised essential K13 compartment proteins [29]. These findings indicate that there are also proteins found at the K13 compartment that have important functions not related to endocytosis or ART resistance.

## KIC12 is located in the nucleus and at the K13 compartment and is involved in endocytosis but not in ART resistance

PF3D7_1329500, currently annotated as 'conserved protein, unknown function' was renamed K̲13 i̲nteracting c̲andidate 12 (KIC12). In order to test whether KIC12 is a member of the K13 compartment or not, we first generated transgenic parasites expressing C-terminally 2xFKBP-GFP-2xFKBP tagged KIC12 from its original genomic locus (KIC12-2xFKBP-GFP-2xFKBP[endo] parasites) (S1E Fig). Expression and localisation of tagged KIC12 was analysed by fluorescent microscopy. KIC12 was detectable in the nucleus in ring stage parasites. In trophozoites foci in the parasite periphery (Fig 3A, white arrows) were observed in addition to the signal in the nucleus (Fig 3A, light blue arrows). In schizonts these foci were not present anymore. Instead, only the nuclear signal and a faint uniform cytoplasmic GFP signal was detected in late trophozoites and early schizonts and these signals were absent in later schizonts and merozoites (Figs 3A, S4A and S4B). In line with this expression pattern, the *kic12* transcriptional profile indicated mRNA levels peak in merozoites and early ring-stage parasites and no RNA expression in trophozoites and schizonts [74,75,82]. Next, we generated KIC12-2xFKBP-GFP-2xFKBP[endo] parasites episomally co-expressing mCherry-K13 which revealed 49% of parasites with overlap, 46% with partial overlap, while 5% showed no overlap of the KIC12 foci with the K13 foci in early trophozoites (Figs 3B and S4A). In rings only 4% of parasites showed an overlap of KIC12 with K13 foci, while in late trophozoites no overlap was observed, in agreement with the exclusively nuclear localisation of KIC12 in these stages (Figs 3B and S4A). We conclude that KIC12 is a protein with a dual localisation in the nucleus and the K13 compartment in trophozoites.

In order to assess the importance of the K13-located pool of KIC12 for parasite proliferation, we generated KIC12-2xFKBP-GFP-2xFKBP[endo] parasites enabling conditional KIC12 inactivation by knock-sideways using a nuclear (1xNLS-FRB-mCherry) mislocaliser. Efficient mislocalisation of KIC12-2xFKBP-GFP-2xFKBP into the nucleus and absence of KIC12 foci in the cytoplasm of trophozoites upon addition of rapalog was confirmed by microscopy at 4 and 16 hours post induction (Fig 3C). Assessing parasite proliferation after knock sideways of

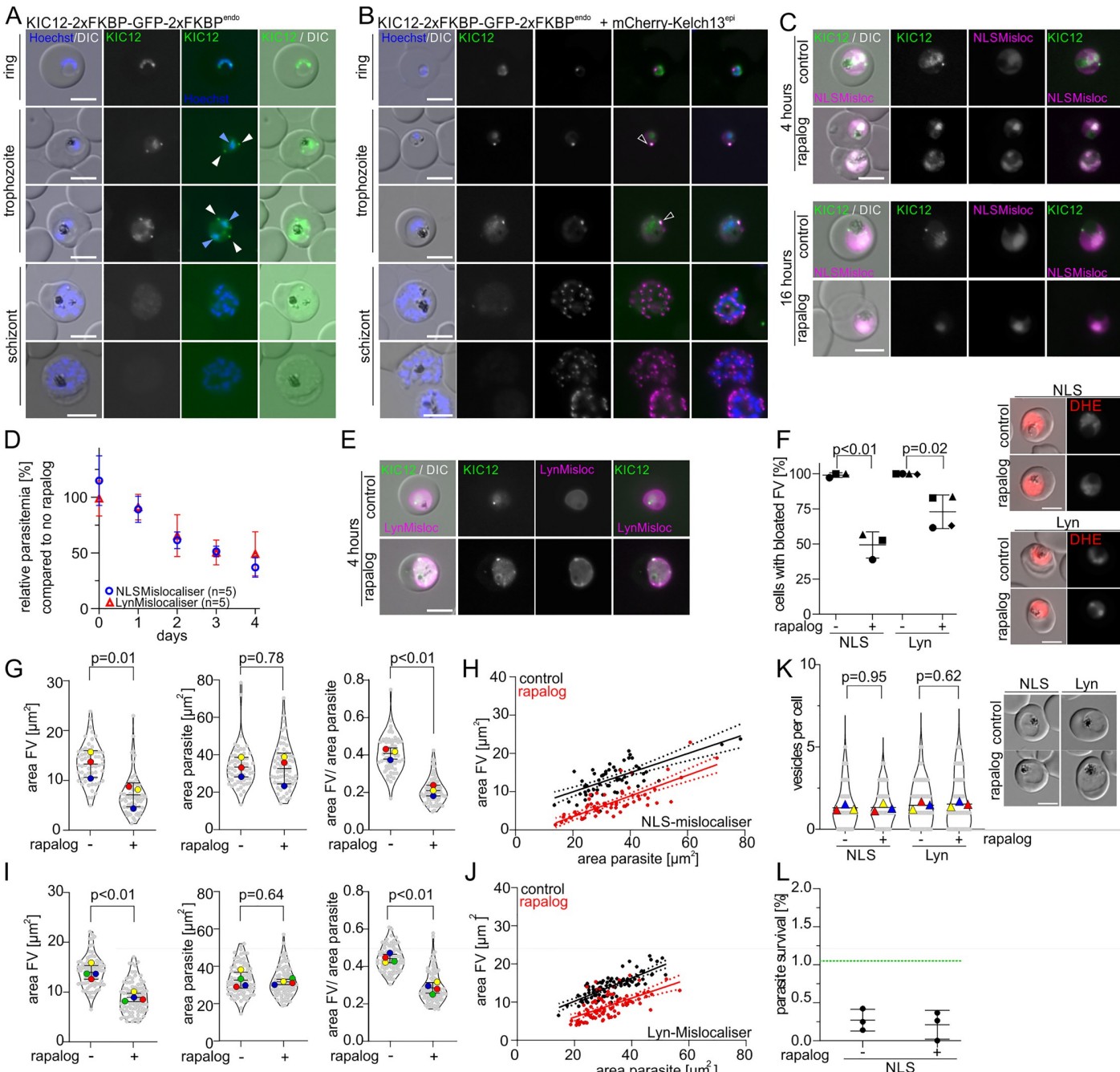

**Fig 3. KIC12 shows a dual localisation in the nucleus and at the K13 compartment and is involved in endocytosis but not in ART resistance. (A)** Localisation of KIC12-2xFKBP-GFP-2xFKBP expressed from the endogenous locus by live-cell microscopy across the intra-erythrocytic development cycle. Arrow heads indicate foci (white) and nuclear (light blue) signal. Nuclei were stained with Hoechst. Scale bar, 5 µm. **(B)** Live cell microscopy images of parasites expressing KIC12-2xFKBP-GFP-2xFKBP with episomally expressed mCherry-K13. Scale bar, 5 µm. Black arrowheads indicate overlapping foci. Nuclei were stained with DAPI. Scale bar, 5 µm. Extended panel shown in S4A Fig. **(C)** Live-cell microscopy of knock sideways (+ rapalog) and control (without rapalog) KIC12-2xFKBP-GFP-2xFKBP^endo+1xNLSmislocaliser parasites 4 and 16 hours after the induction of knock-sideways by addition of rapalog. Scale bar, 5 µm. **(D)** Relative growth of asynchronous KIC12-2xFKBP-GFP-2xFKBP^endo+1xNLSmislocaliser (blue) and KIC12-2xFKBP-GFP-2xFKBP^endo+Lyn-mislocaliser (red) parasites plus rapalog compared with the corresponding untreated control parasites over five days. Five independent growth experiments were performed and mean relative parasitemia +/- SD is shown (all replicas shown in S4C and S4D Fig). **(E)** Live-cell microscopy of knock sideways (+ rapalog) and control (without rapalog) KIC12-2xFKBP-GFP-2xFKBP^endo+Lyn-mislocaliser parasites 4 hours after the induction of knock-sideways by addition of rapalog. Scale bar, 5 µm. **(F)** Bloated food vacuole assay with KIC12-2xFKBP-GFP-2xFKBP^endo parasites 8 hours after inactivation of KIC12 (+rapalog) by NLS-mislocaliser or Lyn-mislocaliser compared with corresponding control (without rapalog). Cells were categorized as 'bloated FV' and 'non-bloated FV'. Results are displayed as percentage of cells with bloated FV. n = 3 (NLS) or n = 4 (LYN) independent experiments were performed with each n = 19–80 (mean 41.3) parasites analysed per condition. P-values determined by Welch's t-test.

Representative DIC and fluorescence microscopy images are shown in the right panel. Parasite cytoplasm was visualized with DHE. Experimental setup shown in S4E Fig. Scale bar, 5 μm. **(G/I)** Area of the FV, area of the parasite and area of FV divided by area of the corresponding parasite of the FV of KIC12-2xFKBP-GFP-2xFKBP$^{endo}$+1xNLSmislocaliser (G) and KIC12-2xFKBP-GFP-2xFKBP$^{endo}$+LYNmislocaliser (I) parasites analysed in 4F. Mean of each independent experiment indicated by coloured dots, individual data points by grey dots. Data presented according to SuperPlot guidelines [147]; error bars represent mean ± SD. P-value determined by paired t-test. **(H/J)** Area of FV of individual cells plotted versus the area of the corresponding parasite in KIC12-2xFKBP-GFP-2xFKBP$^{endo}$+1xNLSmislocaliser and KIC12-2xFKBP-GFP-2xFKBP$^{endo}$+Lyn-mislocaliser parasites of the experiments shown in (F,G and I). Line represents linear regression with error indicated by dashed line. Ten representative DIC images of each independent experiment are shown in S4F and S4G Fig. **(K)** Number of vesicles per parasite in trophozoites determined by live-cell fluorescence microscopy (DIC) in KIC12-2xFKBP-GFP-2xFKBP$^{endo}$+1xNLSmislocaliser and KIC12-2xFKBP-GFP-2xFKBP$^{endo}$+Lyn-mislocaliser parasites with and without rapalog addition. Three independent experiments with n = 10–56 (mean 28) parasites analysed per condition and experiment. Data presented according to SuperPlot guidelines [147]; mean of each independent experiment indicated by coloured triangle, individual data points by grey dots. Error bars represent mean ± SD. P-value determined by paired t-test. Representative DIC images are displayed. **(L)** Parasite survival rate of KIC12-2xFKBP-GFP-2xFKBP$^{endo}$+1xNLSmislocaliser (% survival compared to control without DHA) 66 h after 6 h DHA treatment in standard RSA. Three independent experiments, P-value determined by paired t-test. Green dashed line indicates 1% ART resistance cut-off [26]. 3151–4273 (mean 3690) cells for control and 6209–18941 (mean 12290) cells for DHA treated samples were counted. Experimental setup shown in S4H Fig.

KIC12 showed a mean relative growth of 37.0% compared to control parasites after two development cycles, indicating an important function of the K13 compartment located pool of KIC12 for asexual parasite proliferation (Figs 3D and S4C).

Due to the dual localisation of KIC12 we also generated KIC12-2xFKBP-GFP-2xFKBP$^{endo}$ parasites episomally co-expressing an alternative mislocaliser (Lyn-FRB-mCherry) [73], enabling conditional inactivation of the nuclear pool of KIC12-2xFKBP-GFP-2xFKBP by mislocalisation to the parasite plasma membrane (PPM), an approach previously shown to be suitable for efficient inactivation of nuclear proteins [73,83,84]. Induction of KIC12 mislocalisation to the PPM resulted in a loss of KIC12 in the nucleus 4 hours post induction (Fig 3E). Foci were still detected in the parasite periphery and it is unclear whether these remained with the K13 compartment or were also in some way affected by the Lyn-mislocaliser. Parasite proliferation assays revealed a growth defect of 50.8%, compared with a defect of 63.0% of the nuclear mislocalisation approach (Figs 3D and S4D). Due to the possible influence on the K13 compartment located foci of KIC12 with the Lyn mislocaliser, a clear interpretation in regards to the functional importance of the nuclear pool of KIC12 other than that it confirms the importance of this protein for asexual blood stages is not possible. In contrast, the results with the nuclear mislocaliser indicate that the K13 located pool of KIC12 is important for efficient parasite growth. The general importance of KIC12 for asexual parasite proliferation was further supported by failure to obtain a cell line with a truncated *kic12* using the SLI-TGD system.

Based on the presence at the K13 compartment in trophozoites, we tested the effect of KIC12 inactivation on endocytosis. Bloated food vacuole assays showed that >99% of control parasites developed bloated food vacuoles, while only 49,3% (NLS mislocaliser) or 72.9% (Lyn mislocaliser) of the cells with inactivated KIC12 showed a bloated food vacuole (Figs 3F and S4E), indicating an effect on endocytosis of host cell cytosol. As the effect was only partial, we decided to measure the parasite and food vacuole size. This analysis with the KIC12 NLS mislocaliser parasites revealed a significantly reduced food vacuole size in the parasites with inactivated KIC12, while there was no effect on parasite size (Figs 3G and S4F). Plotting the values of the individual parasites showed that the food vacuoles of similarly sized parasites were consistently smaller in the KIC12 NLS knock sideways compared to controls (Fig 3H), indicating that the effect on hemoglobin delivery to the food vacuole upon inactivation of KIC12 was not an indirect effect due to parasite growth impairment during the assay time. Similar results were obtained using the KIC12 Lyn mislocaliser line (Figs 3I, 3J and S4G). Quantification of the number of vesicles in trophozoites upon inactivation (with either the NLS or Lyn mislocaliser) of KIC12 revealed no difference compared to control (Fig 3K), indicating that KIC12

acted in the early phase of endocytosis, similar to the previously studied K13 compartment proteins [29]. While again the data with the Lyn mislocaliser is more difficult to interpret, the results with the NLS mislocaliser indicate that the K13 compartment located pool of KIC12 is important for endocytosis of host cell cytosol.

The lacking evidence for a K13 compartment localisation of KIC12 in rings indicates that KS using the NLS mislocaliser would not affect ART susceptibility. Indeed, rapalog-induction of the KIC12 NLS mislocaliser KS line showed no significantly decreased ART susceptibility in RSAs (Figs 3L and S4H), congruent with the lacking co-localisation of KIC12 with K13 in rings.

Overall, our results indicate the presence of KIC12 at the K13 compartment in trophozoites, a role in HCCU and an additional pool of KIC12 in the nucleus.

## MCA2 is part of the K13 compartment and its truncation confers *in vitro* ART resistance

We previously identified metacaspase-2 (MCA2) as one of the top 5 most enriched proteins of the K13 proxiome, but it was so far not localised, as endogenous C-terminal tagging with a 2xFKBP-GFP-2xFKBP tag had not been achieved despite several attempts [29]. By screening of the MalariaGEN *Plasmodium falciparum* Community Project database [85] a single-nucleotide polymorphism (SNP) at amino acid position 1344 of MCA2 was identified that on its own leads to a stop codon, thereby removing the metacaspase (MCA) domain. According to MalariaGEN, this SNP is found in Africa with a mean prevalence of 52% and in South East Asia with 5% prevalence [85]. As a previously generated truncation of MCA2 at amino acid position 57 (MCA2[TGD]-GFP (targeted gene disruption (TGD)) reduced the parasites' ART susceptibility as judged by RSAs [29], we generated a second cell line with MCA2 disrupted at the Y1344 position (MCA2[Y1344Stop]-GFP[endo] parasites) (Fig 4A), originally to assess if this change could mediate resistance in the field. Subsequent analyses of African parasite isolates [86] revealed that this change was always accompanied by a second change in the following base, reverting the stop codon, indicating that this change is not of relevance for resistance in endemic settings. However, as we had not so far been able to localise MCA2, we took advantage of these parasites with the first 1344 amino acids of MCA2 (66.5% of the protein) fused to GFP to study its localisation. Correct integration of the construct into the *mca2* genomic locus was confirmed by PCR (S1F Fig) and expression of the truncated fusion protein was verified by Western blot (S1F Fig). Live cell imaging of MCA2[Y1344Stop]-GFP[endo] parasites revealed expression of MCA2 throughout the intraerythrocytic development cycle, appearing as a single focus in ring stage parasites while two or more foci were detectable in trophozoites and schizonts (Fig 4B), in contrast to the predominantly cytosolic localisation of the severely truncated MCA2[TGD]-GFP (S5A Fig). As this localisation is reminiscent of K13 and KIC localisation [29,73] we generated MCA2[Y1344Stop]-GFP[endo] parasite episomally co-expressing mCherry-K13 and the spatial relationship of MCA2[Y1344Stop]-GFP and K13 foci was quantified in ring and trophozoite stage parasites based on fluorescence microscopy (Fig 4C). This analysis showed that 59% of the MCA2[Y1344Stop] foci overlapped with K13 foci, 21% of the foci showed partial overlap and 20% of the MCA2[Y1344Stop] foci did not overlap with K13 foci (Fig 4C), indicating a presence of MCA2 at the K13 compartment. In order to exclude an effect of the truncation on MCA2 localisation we generated parasites with an endogenously 3xHA-tagged full length MCA2 (MCA2-3xHA[endo]) (S1G Fig) that episomally co-expressed mCherry-K13 and performed immunofluorescence assays (IFA). These experiments confirmed the focal localisation of MCA2. Quantification of the relative localisation of MCA2 and K13 revealed that 44% of MCA2 foci overlapped with K13 (S5B Fig). Partial overlap was observed for 24% of MCA2

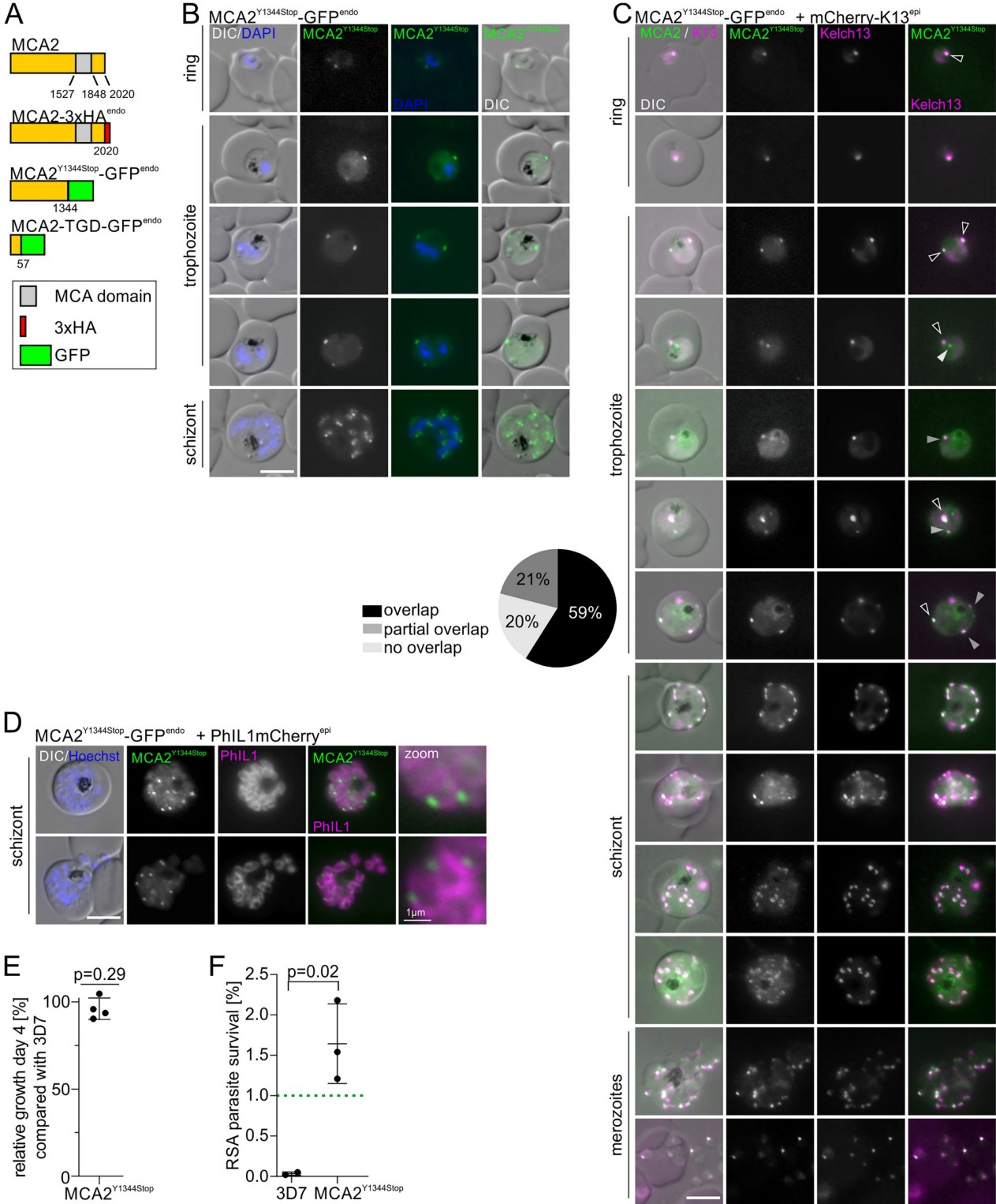

**Fig 4. MCA2 is part of the K13 compartment and its truncation reduces ART susceptibility. (A)** Schematic representation and nomenclature of MCA2 fusion proteins (not to scale) expressed in the parasite by modification of the endogenous gene. GFP in green, 3xHA in red and metacaspase domain in grey. Amino acid positions are indicated by numbers. **(B)** Localisation of MCA2^Y1344STOP-GFP by live-cell microscopy across the intra-erythrocytic development cycle. Nuclei were stained with DAPI. Scale bar, 5 μm. **(C)** Live cell microscopy images of parasites expressing the truncated MCA2^Y1344STOP-GFP fusion protein with an episomally expressed mCherry-K13 fusion protein. Foci from rings and trophozoite stage parasites were

categorized into 'overlap' (black), 'partial overlap' (dark grey) and 'no overlap' (light grey) and shown as frequencies in the pie chart (n = 46 cells were scored from a total of three independent experiments). Scale bar, 5 µm. **(D)** Live cell microscopy images of parasites expressing the truncated MCA2Y1344STOP-GFP fusion protein with the IMC marker protein PhIL1mCherry. Zoom, enlarged region (factor 400%). Nuclei were stained with Hoechst-33342; scale bars, 5 µm and for zoom 1 µm. **(E)** Relative growth of synchronised MCA2Y1344STOP compared with 3D7 wild type parasites after two growth cycles. Each dot shows one of four independent experiments. P-values determined by one-sample t-test. **(F)** Parasite survival rate (% survival compared with control without DHA) 66 h after 6 h DHA treatment in standard RSA. Two (3D7) or three (MCA2Y1344STOP) independent experiments, P-value determined by unpaired t-test. Green dashed line indicates 1% ART resistance cut-off [26]; 3419–3828 (mean 3632) erythrocytes for control and 10364–11734 (mean 11254) cells for DHA treated samples were counted.

foci, whereas 32% MCA2 foci did not overlap with K13 (S5B Fig). Overall, these findings indicated that MCA2 is a K13 compartment protein but is also found in additional, non-K13 compartment foci.

MCA2 was recently found to be enriched in BioID experiments with the inner membrane complex (IMC) marker protein PhIL1 [65]. We generated MCA2Y1344Stop-GFPendo parasites episomally co-expressing PhIL1mCherry and analysed schizont stage parasites. Fluorescent imaging with these parasites revealed a close association of MCA2Y1344Stop-GFP foci with the IMC at the periphery of the newly formed merozoites (Fig 4D), as previously observed for K13 [65]. This data indicated that MCA2 and the K13 compartment—as previously suggested [30,65]—are found proximal to the IMC in schizonts.

We previously showed that a truncation of MCA2 at amino acid position 57 (2.8% of the protein left) results in significantly reduced *in vitro* parasite proliferation [29]. Proliferation assays with the MCA2Y1344Stop-GFPendo parasites which express a much larger portion of this protein (66.5% of the protein left), yet still lack the MCA domain (Fig 4A), indicated no growth defect in these parasites compared to 3D7 wild type parasites (Fig 4E). Hence, the MCA domain in MCA2 does not appear to be needed for efficient *in vitro* blood stage growth.

Next, we tested the effect of MCA2 truncation on endocytosis. For this we adapted the bloated food vacuole assays to compare MCA2-TGD and MCA2Y1344Stop-GFPendo with 3D7 parasites. The assay showed that in all three cell lines >95% of parasites developed bloated food vacuoles (S5C Fig). Additionally, there was no apparent difference in the size of food vacuoles of similarly sized parasites between the three lines (S5D and S5E Fig). Of note, in line with the reduced parasite proliferation rate [29], an effect on parasite and food vacuole size was observed for the MCA2-TGD parasites (S5D and S5E Fig). Overall, with this assay, there was no detectable effect on endocytosis in trophozoites in these parasite lines.

Last, we performed RSAs which revealed that the truncation of MCA2 at amino acid position Y1344 resulted in a reduced sensitivity to ART (Fig 4F). The mean parasite survival rate was 1.64%, which is above the defined ART resistance cut-off value of 1% [26], but lower than the ~5% survival rate of the MCA2-TGD parasites [29]. This considerable reduction in ART susceptibility in the parasites with the truncation at MCA2 position 57 compared to the parasites still expressing 1344 amino acids of MCA2, despite both versions of the protein lacking the MCA domain, indicates that the influence on ART resistance is not, or only partially due to the MCA domain.

## Candidate proteins not detected at the K13 compartment

We also generated parasites expressing endogenously C-terminally 2xFKBP-GFP-2xFKBP tagged UIS14, PF3D7_1365800, PF3D7_1447800, PF3D7_0907200 and VPS51 and episomally co-expressed mCherry-K13 (S1H and S1L Fig). Fluorescence microscopy revealed no clear association with the K13 compartment in rings and trophozoites in any of these parasite lines. Instead UIS14, PF3D7_1447800 and VPS51 showed GFP foci in the parasite cytosol without

consistent overlap with mCherry-K13 foci (S6G, S7A and S7B Figs), PF3D7_0907200 showed a weak cytosolic GFP signal and PF3D7_1365800 showed cytosolic GFP signal with additional foci closely associated with the nucleus without consistent overlap of the main foci with mCherry-K13 foci (S6G, S6F, S7A and S7B Figs). Several attempts to generate PF3D7_1243400-2xFKBP-GFP-2xFKBP[endo] or 2xFKBP-GFP-2xFKBP-PF3D7_1243400[endo] parasites remained unsuccessful, indicating the gene might be refractory to C- and N-terminal modification and hence might be essential. For *vps51* and *uis14* we additionally were able to generate targeted gene disruption cell lines (S1M, S1N, S6B and S6E Figs), indicating these candidates are dispensable for *in vitro* asexual parasite proliferation, although growth assays indicated a need of UIS14 for optimal growth (S6C Fig).

Structural homology search [87] revealed the presence of an N-terminal arfaptin homology (AH) domain in PF3D7_1365800 (S8C Fig), a domain known to promote binding of arfaptin to Arf and Rho family GTPases important for vesicle budding at the Golgi [88,89]. Given that in *Toxoplasma* an intersection of endocytosis and secretion was observed at the trans Golgi [90], we tested the potential (indirect) influence of this protein on endocytosis-related processes. We performed conditional inactivation using knock-sideways, but despite efficient loss of the PF3D7_1365800 foci, no growth defect was observed (S7C and S7D Fig) and co-expression of the Golgi marker GRASP [91] revealed no consistent overlap between the foci of these two proteins (S7E Fig).

Based on this analysis we did not classify UIS14, PF3D7_1365800, PF3D7_1447800, PF3D7_0907200, and VPS51 as K13 compartment proteins and the location of PF3D7_1243400 remains unknown.

## The domain repertoire in K13-compartment proteins

With the extended complement of K13 compartment proteins from this and previous work [29], we assessed the repertoire of functional domains at this site. For this we took advantage of recent advances in protein structure prediction to identify structural similarities in K13-compartment members for which no information could be inferred from sequence homology. We compared their structures predicted with the AlphaFold algorithm [87,92] with experimentally determined protein structures in the Protein Data Bank and identified 25 domains, 15 of which were not previously identified according to PlasmoDB and Interpro (Figs 5A, 5B and S8).

The largest number of recognisable folds were detected in KIC4, a protein for which we previously detected some similarity to α-adaptins [29]. KIC4 contained an N-terminal VHS domain (IPR002014), followed by a GAT domain (IPR004152) and an Ig-like clathrin adaptor α/β/γ adaptin appendage domain (IPR008152) (Figs 5A, 5C and S8). This is an arrangement typical for GGAs (Golgi-localised gamma ear-containing Arf-binding proteins) which are vesicle adaptors first found to function at the trans-Golgi [93,94]. Surprisingly, KIC4 however also contains an additional domain at its C-terminus, a β-adaptin appendage C-terminal subdomain (IPR015151) which is a part of the ear domain of β-adaptins and not found in GGAs. Together with the preceding clathrin adaptor α/β/γ adaptin domain, the C-terminus of KIC4 therefore resembles a β-adaptin. The region without detectable fold after the GAT domain likely corresponds to the hinge region in GGAs. This suggests that KIC4 is a hybrid between GGAs and an AP complex subunit beta (Fig 5C), the two known types of adaptors. Based on Interpro [95], such a domain organization has to date not been observed in any other protein.

KIC5 also contains a clathrin adaptor α/β/γ-adaptin domain (IPR008152) and one of the two subdomains of the ear domain of α-adaptins (clathrin adaptor α-adaptin_appendage C-terminal subdomain, IPR003164) (Fig 5A, 5B and 5D). More than 97% of proteins containing

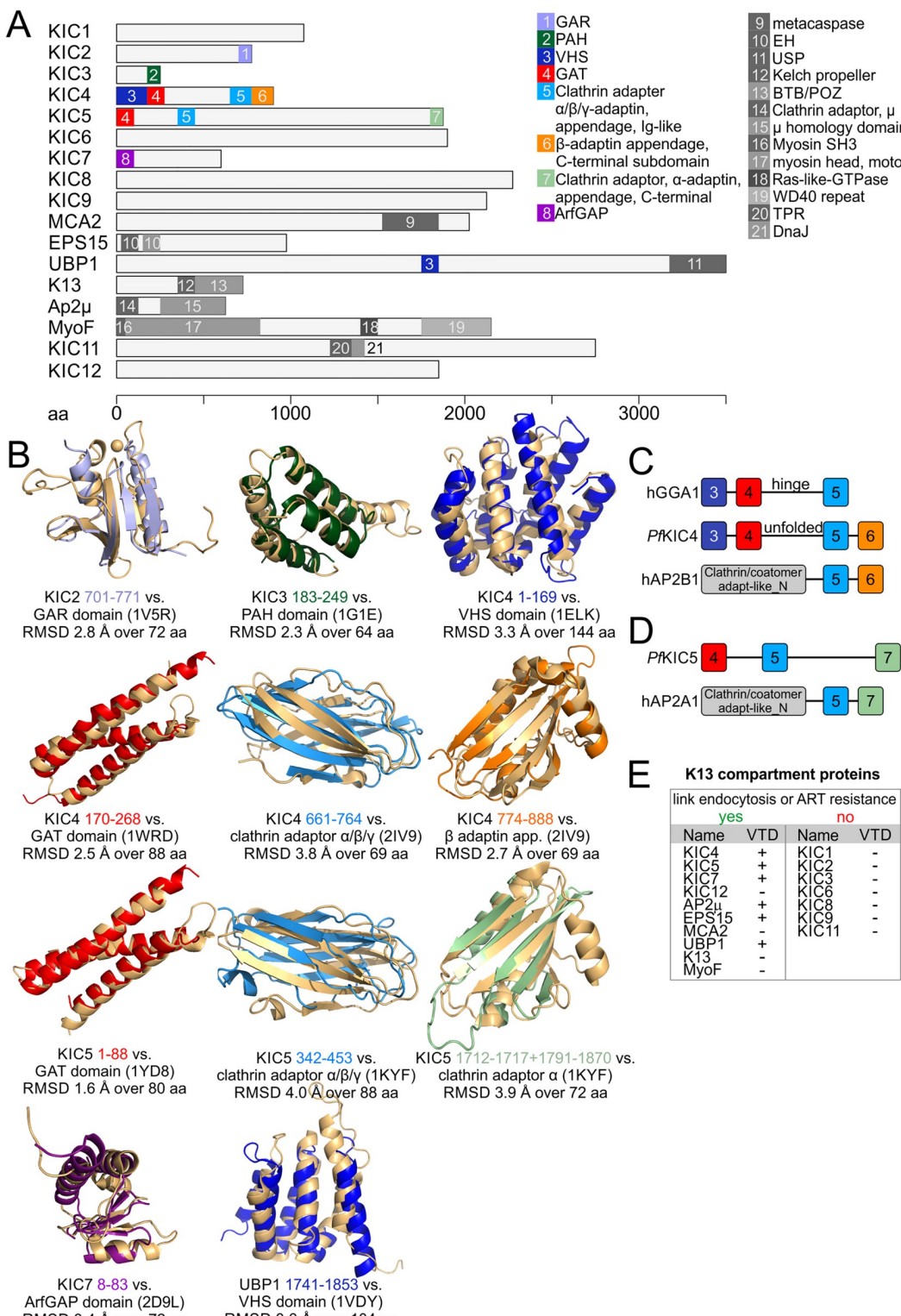

**Fig 5. Type of domain found in K13 compartment proteins coincide with functional group. (A)** K13-compartment members shown to scale with domains indicated that were identified from AlphaFold structure predictions. Newly identified domains (i.e. domains not previously identified by sequence homology) are shown in colours, previously known domains are shown in grey. A brief summary of the function of each newly identified domain and Interpro IDs of all domains can be found in S8 Fig. **(B)** Alphafold prediction of each newly identified domain is shown in the same colour as in A and aligned

with the most similar domain from the PDB. PDB ID and alignment details are indicated beneath each set of aligned domains. Root mean square deviations (RMSD) for all comparisons are given. (**C**) Domain organizations of human GGA1 (hGGA1), KIC4 and human AP-2 subunit beta 1 (hAP2B1) are shown (not to scale). Domains are coloured as in A and B. (**D**) Domain organizations of KIC5 and human AP-2 subunit alpha 1 (hAP2A1) are shown (not to scale). Domains are coloured as in A, B and C. (**E**) Table summarising K13 compartment proteins according to property (presence of vesicle trafficking domain (VTD)) and function.

these domains also contain an Adaptin_N (IPR002553) domain [95] and in this combination typically function in vesicle adaptor complexes as subunit α [96,97] (Fig 5D) but no such domain was detectable in KIC5. KIC5 thus displays some similarities to AP complex subunit α but similar to KIC4, there appear to be profound difference to the canonical adaptors.

Based on the identified adaptor domains in KIC4 and KIC5 we decided to test for an effect on endocytosis in previously established parasites with truncations in these genes [29]. But neither KIC4-TGD nor KIC5-TGD parasites showed a difference in bloated food vacuole assay compared to 3D7 parasites and knock sideways of KIC4, in agreement with absence of a growth defect in the TGD, did not affect growth nor endocytosis (S10 Fig). Of note, as for MCA2-TGD, a reduced parasite and food vacuole size was observed for KIC5-TGD, which is in line with the reported reduced parasite proliferation [29]. While this overall indicates no role of these proteins in endocytosis in trophozoites, we note that TGD lines might not be ideal to capture non-absolute effects on endocytosis. Also, if an endocytosis defect were the reason for the slowed parasite growth in the KIC5- and MCA2-TGD lines, our assay would not show this as an endocytosis defect.

KIC7 contains an ArfGAP domain, as recently also predicted for its *Toxoplasma* homolog AGFG (TGME49_257070) [98]. ArfGAPs regulate the activity of Arfs which are small GTP binding proteins coordinating vesicle trafficking [99,100]. UBP1 contains a ubiquitin specific protease (USP) domain at its C-terminus which previously led to its name ubiquitin carboxyl-terminal hydrolase 1 (UBP1) [101]. Here we also identified a VHS domain in its centre. VHS domains occur in vesicular trafficking proteins, particularly in endocytosis proteins, but typically are found at the N-terminus of proteins (in over 99.8% of cases according to annotations by Interpro). A combination with a USP domain has not been observed so far. If the VHS domain is functional in UBP1 despite its central position, it is the first structural domain that would support the functional data showing this K13 compartment protein has a role in endocytosis [29].

KIC2 contains a GAR domain which typically binds the cytoskeleton [102] and KIC3 contains a PAH domain which can serve as a scaffold for transcription factors [103,104]. While not consistently detected at the K13 compartment, it is interesting that we here found that PF3D7_1365800 contains an AH domain, a member of the AH/BAR domain family (S8C Fig). According to Interpro, no AH or BAR domain proteins have so far been detected in malaria parasites, rendering this an interesting protein. However, we found this protein as being likely dispensable for intra-erythrocytic parasite development and no colocalisation with K13 was observed, suggesting PF3D7_1365800 is not needed for endocytosis.

Overall, this analysis revealed that most of the proteins involved in endocytosis or *in vitro* ART resistance contain regions with structural homology to vesicle trafficking domains, often specific for endocytosis (Fig 5E). However, apart from AP2 the domain arrangements are unusual, the conservation on the primary sequence level is low (which precluded initial detection) and there are large regions without any resemblance to other proteins, altogether indicating strong parasite-specific adaptations. In contrast, the K13 compartment proteins where no role in ART resistance (based on RSA) or endocytosis was detected, KIC1, KIC2, KIC6, KIC8,

KIC9 and KIC11, do not contain such domains (Fig 5E). This second group of proteins might have different functions that—apart from KIC11—seem to be dispensable for blood stage growth.

## Discussion

The BioID proxiome of K13 and Eps15 revealed the first proteins involved in the initial steps of endocytosis in malaria parasites, a process that in blood stage parasites leads to the uptake of large quantities of host cell cytosol. As demonstrated in model organisms, endocytosis is a complex and highly regulated process involving a multitude of proteins [105–107], but in Apicomplexan parasites is not well studied [42]. Thus, the K13 compartment proxiome represents an opportunity to identify proteins involved in this process in malaria parasites and will be important to understand it on a mechanistic level. Understanding how and if HCCU differs from the canonical endocytic processes in human cells, will not only help to understand this critical process in parasite biology but might also reveal parasite-specific aspects that permit specific inhibition and could be targets for drug development efforts.

Here we expanded the repertoire of K13 compartment proteins and functionally analysed several of them. An assessment of structural similarities indicated an abundance of vesicle trafficking domains—often typical for endocytosis—in the confirmed K13 compartment proteins. A similar search with all un-annotated *P. falciparum* proteins (in total more than 900 proteins) detected only 7 typical vesicle trafficking domains [108]. While this did not include annotated proteins, it nevertheless indicates that vesicle trafficking domains are overrepresented in the proteins found at the K13 compartment. The proteins with vesicle trafficking domains at the K13 compartment now comprise KIC4, KIC5, KIC7, Eps15, AP2μ and UBP1, all of which reduce *in vitro* ART susceptibility when inactivated, suggesting that this led to a reduced endocytosis in ring stages [29]. A role in endocytosis has been experimentally shown for all of these except for KIC4 and KIC5, which are both non-essential for the parasites asexual blood stage development and for which gene disruptions were available [29]. We here nevertheless attempted to assess endocytosis in these and the MCA2 TGD line as well as in a KIC4 knock sideways line but failed to detect an effect. For KIC4 this was not surprising, as it is dispensable, speaking against an important role in endocytosis. This indicates that the elevated RSA survival in the KIC4-TGD parasites is due to other reasons than endocytosis or that this TGD leads to a reduction in endocytosis in rings only (like Kelch13 [29]), or that the endocytosis reduction is too small to detect in our assay. The latter is possible because a comparably small reduction in protein abundance (or activity) of endocytosis proteins can already lead to a substantial increase in RSA survival [29,40] and the KIC4 TGD only has a comparably small increased in RSA survival to just above the 1% threshold [29]. In contrast to KIC4, the KIC5 and MCA2 TGD lines both have a growth defect. We note that if the growth defect in the TGD lines is due to endocytosis, the food vacuole size reduction may be proportional to parasite size reduction which would not be classified as an endocytosis defect in our assay. It is also possible that these proteins are only important for endocytosis in ring stages. We conclude that either these proteins influence ART susceptibility through a different function than endocytosis or that technical limitations or stage-specificity led to a failure to detect a contribution of these proteins to endocytosis. Further work will be needed to clarify this.

With KIC4 and KIC5 malaria parasites contain two additional K13 compartment proteins (besides the AP2 complex) that based on our structural similarity analysis contain domains found in adaptor subunits. KIC4 and KIC5 likely have additional rather than redundant roles to the AP2 complex, as the AP2 complex is essential for blood stage growth and on its own is needed for HCCU [29,109]. While KIC4 disruption did not lead to a growth defect, disruption

of KIC5 impaired parasite growth [29], indicating that there cannot be full redundancy between these two proteins either. Of note, in *T. gondii* a homologue of KIC4 (*Tg*KAE, TGME49_272600) was detected at the micropore and shown to be essential, while no homologue of KIC5 was identified [98,110]. Altogether, these considerations support specific individual roles of the known K13 compartment proteins with adaptor domains. Recent work indicated a link between KIC5, nuclear metabolism and ART sensitivity [111]. We did not detect KIC5 in the nucleus [29]. Further work is needed to understand how KIC5 influences these functions.

In model organisms, the α/β/γ-adaptin appendage domain, the α-adaptin domain and the β-adaptin domain, in the AP2 complex and in GGAs, act as interaction hubs for more than 15 accessory proteins, including Eps15, Arfs, amphiphysin, epsins and rabaptins as well as lipids involved in vesicle budding [112]. This fits with the presence of Eps15 and KIC7 (which contains an ArfGAP domain) at the K13 compartment. The interaction between the α/β/γ-adaptin appendage domain and Eps15 has been captured by X-ray crystallography (2I9V) [112], highlighting a possible functional connection between Eps15 and AP-2α, AP-2β, KIC4 or KIC5, which all contain this domain. Indeed, the *Toxoplasma* homologue of KIC4 has recently been shown to bind Eps15, while *Tg*AP-2α did not [98].

Despite the detection of various domains in K13 compartment proteins it is noteworthy that most of these proteins (e.g. EPS15, UBP1, KIC7) still contain large regions without any homology to other proteins. The parasite-specific nature of the initial steps of endocytosis is also evident from the difficulty of primary sequence-based detection of the vesicular trafficking domains in K13 compartment proteins and from the unusual domain combinations. Furthermore it is peculiar that despite the presence of the clathrin adaptor AP2, clathrin itself does not seem to be involved [29,109], further indicating parasite-specific features of HCCU. Overall, this indicates a strongly adapted mechanism of the first steps in endocytosis for HCCU in malaria parasites. One protein typically involved in endocytosis that did not appear in the list of highly enriched proteins of the K13 and Eps15 proxiome is a dynamin [113] which might indicate further differences to the canonical mechanism in model organism. However, at least in *Toxoplasma* an association of K13 and its compartment with a dynamin was reported [98,114], indicating that an equivalent is likely also present in malaria parasites.

A protein that has a more canonical structure is MyoF, a Class XXII myosin [77], which we here link with endocytosis and found in foci that were often close or overlapping with the K13 compartment. The only partial overlap could indicate that either it i) only transiently associates with the K13 compartment, or ii) is in a separate compartment that is close to the K13 compartment or iii) is in a region of the K13 compartment that is close, but non-overlapping, with that defined by the other so far known K13 compartment proteins. A number of conclusions can be drawn from our MyoF characterisation. Firstly, its inactivation resulted in the appearance of vesicles, similar to the phenotype of VPS45 inactivation [54], indicating it has a role in endosomal transport, downstream of the initial steps of endocytosis. This is in contrast to the other K13 compartment proteins and might explain the only partial overlap in location of MyoF with the K13 compartment. Secondly, its involvement suggests a role of actin/myosin in endosomal transport which is well known from other organisms [115,116] and supports the observation that the actin inhibitor CytochalasinD leads to vesicles similar to MyoF inactivation [53]. Hence, myosin may generate force needed for the transport of host cell cytosol filled vesicles to the food vacuole. Thirdly, there was no effect of MyoF inactivation or CytochalasinD treatment on ART resistance, which might indicate that there is no need for actin/myosin for endocytosis in rings. Although there were some limitations to our system to study MyoF, the substantial inactivation caused by simply tagging MyoF would already have led to a decreased susceptibility to ART in RSAs (as seen with other K13 compartment proteins [29]),

if ring stage endocytosis had been affected, which was not the case. It is of note that the endocytosis function of MyoF is reciprocal to that of K13 which is only required in rings, indicating that there appear to be stage-specific differences in endocytosis between these stages. However, the majority of components are at the K13 compartment throughout the cycle and for instance KIC7 is needed for HCCU in both rings and trophozoites [29]. Of note, mutations in *myoF* have previously been found to be associated with reduced ART susceptibility [117], but 12 mutations tested in the laboratory strain 3D7 did not result in increased RSA survival [40].

Attempts to generate a knockout of *P. berghei myof* were reported to be unsuccessful [77], while a genome-wide mutagenesis screen predicted *Pf*MyoF to be dispensable for asexual blood stage development [118]. *Tg*MyoF has previously been implicated in Golgi and rhoptry positioning [119], trafficking of dense granule [120] or nuclear encoded apicoplast proteins [121] or centrosomes positioning and apicoplast inheritance [122]. We cannot exclude that beside the effect on HCCU, a part of the growth defect we see after MyoF inactivation arises from similar functions in malaria blood stages.

MCA2 has been identified as putative K13 interaction candidate, but its location was unknown [29]. Here, we establish MCA2 as member of the K13 compartment by endogenously tagging it with HA as well as by using a cell line with a truncated MCA2 fused to GFP. MCA2 was located in foci overlapping with the K13 compartment and in schizonts the foci were in proximity of the IMC in line with previous data linking the K13 compartment and the IMC [65,123]. A close association of the K13 compartment and the IMC is also supported by recent work in *T. gondii* [98,110,124] showing that the K13 compartment and its proteins are present at the IMC embedded micropore [98,110].

We previously showed that gene disruption of *mca2* results in reduced *in vitro* parasite proliferation and *in vitro* ART resistance [29], which is also supported by a knockout of the *mca2* homologue in the rodent infecting *P. berghei* [125,126]. Truncation of MCA2 at amino acid position Y1344 still resulted in decreased ART susceptibility, even though the susceptibility reduction was less than a disruption at amino acid position 57. In contrast to the full disruption, truncation at residue 1344 did not lead to a growth defect, indicating that the predicted metacaspase domain (AA 1527–1848)—not present in this truncated MCA2 protein—is dispensable for asexual parasite proliferation. This finding is in contrast to results indicating an important role of the metacaspase domain for parasite proliferation using MCA2 inhibitors [127–129].

Here we also identified the first protein, KIC11, at the K13 compartment that had an important function for the growth of blood stage parasites but did not appear to function in endocytosis. Instead, KIC11 had an important function in invasion or possibly early ring stage development. It is at present unclear if this function is carried out by KIC11 at the K13 compartment or the KIC11 in the foci not overlapping with K13. KIC11 was predicted to be non-essential by a genome-wide mutagenesis screen [118], while the orthologue in the rodent malaria parasite *P. berghei* (PBANKA_0906900) was classified as essential for asexual proliferation in the PlasmoGem screen [130].

Overall the classification of K13 compartment proteins presented in this work indicates that there are two main groups. The first group includes proteins that define a highly unusual endocytic pathway to internalise host cell cytosol. These proteins are predominately parasite-specific (exceptions being AP2μ, MyoF and in part K13 which however lacks vesicle trafficking domains). This group includes proteins that are involved in *in vitro* ART resistance (K13, MCA2, UBP1, KIC7), but also proteins such as KIC12 or MyoF that do not confer *in vitro* ART resistance when inactivated, presumably because they are not needed for endocytosis in rings. In the case of MyoF this idea is supported by low expression in rings and lacking effect of cytochalasin D in rings whereas KIC12 was not at the K13 compartment in rings. Reciprocal

to this, we found in previous work that K13 is needed for endocytosis in ring stages only [29]. Hence, there is heterogeneity in the stage-specificity of the endocytosis proteins at the K13 compartment. It is also important to note that we failed to detect an endocytosis function (at least in trophozoites) for KIC4, KIC5 and MCA2 which do reduce ART susceptibility when inactivated. It therefore is possible that they influence ART susceptibility through different functions although at least KIC4 and KIC5 contain multiple domains indicative of a function in a vesicle trafficking process. Of the proteins in this group for which an endocytosis function was shown most are involved in the initial phase of HCCU. In contrast to the non-K13 compartment protein VPS45 [54] their inactivation does not result in the generation of endosomal intermediates in the parasite cytoplasm. The exception is MyoF which (of the proteins here classified as K13 compartment associated) generated vesicles when inactivated. MyoF showed the lowest spatial overlap with K13 and may form a link to downstream steps of endosomal transport.

The second group of K13 compartment proteins neither affects RSA survival nor endocytosis nor does it contain vesicle trafficking domains (as based on our domain identification here). These proteins may serve other functions, but as most are dispensable for *in vitro* asexual blood stage growth, their roles and the homogeneity of functions of this group is unclear. KIC11, the first essential protein of this group, might help to shed light on the function of the non-endocytosis related K13 compartment proteins. However, it should be noted that KIC11 showed an additional protein pool independent of the K13 compartment, most prominently in schizonts. Hence, it is possible that KIC11's essential function is not related to the K13 compartment but is carried out by the pool of KIC11 located elsewhere.

Overall, this work strengthens the notion that the K13 compartment is involved in a vesicular trafficking process, reveals novel essential HCCU proteins and provides a classification of its members that might inform future studies to understand the unusual mechanism of endocytosis in apicomplexans.

## Methods

### *P. falciparum* culture

Blood stages of *P. falciparum* 3D7 [131] were cultured in human red blood cells (O+; University Medical Center Hamburg, Eppendorf (UKE)). Cultures were maintained at 37˚C in an atmosphere of 1% $O_2$, 5% $CO_2$ and 94% $N_2$ using RPMI complete medium containing 0.5% Albumax according to standard protocols [132].

In order to obtain highly synchronous parasite cultures, late schizonts were isolated by percoll gradient [133] and cultured with fresh erythrocytes for 4 hours. Afterwards sorbitol synchronization [134] was applied in order to remove remaining schizonts resulting in a highly synchronous ring stage parasite culture with a four-hour age window.

### Cloning of plasmid constructs for parasite transfection

For endogenous C-terminal 2x-FKBP-GFP-2xFKBP tagging using the SLI system [73] a homology region of 321–1044 bp (1044 bp for *Pfmyof* (PF3D7_1329100), 690 bp for *Pfkic11* (PF3D7_1142100), 780 bp for PF3D7_1447800, 695 bp for *Pfkic12* (PF3D7_1329500), 411 bp for PF3D7_1243400, 321 bp for PF3D7_1365800, 674 bp for *Pfuis14* (PF3D7_0405700), 698 bp for PF3D7_0907200, 756 bp for *Pfvps51* (PF3D7_0103100)) was amplified from 3D7 gDNA and cloned into pSLI-sandwich [73] using the NotI/AvrII restriction sites.

For endogenous C-terminal 2x-FKBP-GFP tagging using the SLI system [73] a homology region of 1044 bp for *Pfmyof* (PF3D7_1329100), was amplified using 3D7 gDNA and cloned into pSLI-2xFKBP-GFP [73] using the NotI/AvrII restriction sites.

For endogenous C-terminal 3xHA tagging using the SLI system [73] a homology region of 948–999 bp (999 bp for *Pfmca2* (PF3D7_1438400), 948 bp for *Pfmyof* (PF3D7_1329100)) was amplified using 3D7 gDNA and cloned into pSLI-3xHA [135] using the NotI/XhoI restriction sites.

For endogenous N-terminal 2x-FKBP-GFP-2xFKBP tagging of PF3D7_1243400 using the SLI system [73] a homology region of 276 bp was amplified from 3D7 gDNA and cloned into p-N-SLI-sandwich-loxp [73] using the NotI/PmeI restriction sites. A functional and codon-changed version of PF3D7_1243400 including the C-terminal loxp site was synthesised (Bio-Cat GmbH, Germany) and cloned into the construct using the AvrII/XhoI restriction sites.

For generating *Pf*MCA2^Y1344Stop^-GFP^endo^ a 984 bp homology region was amplified using 3D7 gDNA and cloned into the pSLI-TGD plasmid [73] using NotI and MluI restriction sites.

For targeted gene disruption (TGD) a 429–617 bp (534 bp for *Pfkic12* (PF3D7_1329500), 615 bp for *Pfkic11* (PF3D7_1142100), 617 bp for *Pfuis14* (PF3D7_0405700), 429 bp for *Pfvps51* (PF3D7_0103100)) was amplified using 3D7 gDNA and cloned into the pSLI-TGD plasmid [73] using NotI and MluI restriction sites.

All oligonucleotides used to generate DNA fragments as well as those used for genotyping PCRs are listed in S2 Table.

For co-localisation experiments the plasmids p40PX-mCherry [54], pmCherry-K13_DHODH^nmd3^ [29], ^ama1^PhIL1mCherry [136] and pGRASPmCherry-BSD^nmd3^ [29] were used.

In order to generate the plasmids for co-localisation experiments while simultaneously carrying out knock sideways, ^ama1^AMA1mCherry, ^ama1^AROmCherry and ^ama1^IMC1cmCherry were amplified from pARL^ama1^AMA1mCherry, pARL^ama1^AROmCherry or pARL^ama1^IMC1cmCherry [65] by PCR and cloned into *ama1*-p40-mSca_*nmd3*'-NLS-FRB-T2A-yDHODH, which was generated by inserting the *ama1* promoter using the NotI/KpnI restriction sites into *sf3a2*-p40-mSca_*nmd3*'-NLS-FRB-T2A-yDHODH [55], using the KpnI/XmaI restriction sites, resulting in ^ama1^AROmCherry_*nmd3*'-NLS-FRB-T2A-yDHODH, ^ama1^AMA1mCherry_*nmd3*'-NLS-FRB-T2A-yDHODH and ^ama1^IMC1cmCherry_*nmd3*'-NLS-FRB-T2A-yDHODH.

## Transfection of *P. falciparum*

For transfection, Percoll-purified [133] parasites at late schizont stage were transfected with 50 μg plasmid DNA using Amaxa Nucleofector 2b (Lonza, Switzerland) as previously described [137]. Transfectants were selected using either 4 nM WR99210 (Jacobus Pharmaceuticals), 0.9 μM DSM1 [138] (BEI Resources) or 2 mg/ml for Blasticidin (BSD) (Invitrogen). In order to select for parasites carrying the genomic modification via the SLI system [73], G418 (ThermoFisher, USA) at a final concentration of 400 μg/mL was added to a culture with about 5% parasitemia. The selection process and integration test were performed as previously described [73].

## Imaging

All fluorescence images were captured using a Zeiss Axioskop 2plus microscope with a Hamamatsu Digital camera (Model C4742-95).

Microscopy of live parasite-infected erythrocytes was performed as previously described [139]. Approximately 5 μL of infected erythrocytes were added on a glass slide and covered with a cover slip. Nuclei were stained with 1 μg/mL Hoechst-33342 (Invitrogen) or 1μg/mL 4',6'-diamidine-2'-phenylindole dihydrochloride (DAPI) (Roche).

Immunofluorescence assays (IFA) were performed as previously described [135]. Asynchronous parasite cultures with 5% parasitemia were harvested, washed twice with PBS, air-dried as thin monolayers on 10-well slides (Thermo Fischer) and fixed in 100% acetone for 30 min at room temperature. After rehydration with PBS, the cells were incubated with primary antibody solution containing rabbit a-HA (1:500) (Cell Signalling) and rat a-RFP (1:500) (Chromotek) diluted in 3% BSA in PBS. After three wash steps with PBS, incubation with corresponding secondary antibodies (Molecular probes) was performed. Slides were sealed with a coverslip using mounting medium (Dako).

## Parasite proliferation assay

For parasite proliferation assays a flow cytometry based assay, adapted from previously published assays [54,140], was performed to measure multiplication over five days. These assays were either preformed using synchronous or non-synchronous parasites, as indicated in the corresponding figure legends. For synchronous proliferation assays schizonts were isolated by percoll gradient [133] and incubated with fresh erythrocytes at 37˚C for 4h followed by sorbitol synchronization [134], resulting in ring stage parasite cultures with a 4h time window.

For knock sideways experiments, the same culture was split into control and + 250 nM rapalog at the start of the proliferation assay and growth of each of these two cultures followed over the course of the assay. In order to determine parasitemia, parasite cultures were resuspended and 20 µL samples were transferred to an Eppendorf tube containing 80 µL RPMI. Hoechst-33342 and dihydroethidium (DHE) were added to obtain final concentrations of 5 µg/mL and 4.5 µg/mL, respectively. Samples were incubated for 20 min (protected from UV light) at room temperature, and parasitemia was determined using an LSRII flow cytometer by counting 100,000 events using the FACSDiva software (BD Biosciences).

## Immunoblotting

Protein samples were resolved by SDS-PAGE and transferred to Amersham Protran membranes (GE Healthcare) in a tankblot device (Bio-Rad) using transfer buffer (0.192 M Glycine, 0.1% SDS, 25 mM Tris) with 20% methanol. Next, membranes were blocked for 30 minutes with 5% skim milk, and incubated with primary antibodies diluted in PBS containing 5% skim milk for 2h or overnight, followed by three washing steps with PBS and 2h incubation with horseradish peroxidase-conjugated secondary antibodies diluted in PBS containing 5% skim milk. Detection was performed using the Clarity Western ECL kit (Bio-Rad), and signals were recorded with a ChemiDoc XRS imaging system (Bio-Rad) equipped with Image Lab software 5.2 (Bio-Rad).

Antibodies were applied in the following dilutions: mouse a-GFP (1:1000) (Roche), rat a-HA (1:2000) (Roche), rabbit anti-aldolase (1.2000) [141], goat α-rat (1:2000) (Dianova), goat α-mouse (1:2000) (Dianova) and donkey α-rabbit (1:2000) (Dianova).

## Conditional inactivation via knock-sideways

For knock-sideways cell lines were transfected with plasmids encoding the nuclear mislocaliser (NLS-FRB-mCherry) of the PPM mislocaliser (Lyn-FRB-mCherry) [73]. The knock-sideways approach was performed as described previously [73]. Briefly, cultures were split into two 2-ml cultures of which one was supplemented with 250 nM rapalog (Clontech). Mislocalisation of the target protein was verified by live-cell microscopy.

### Ring stage survival assay (RSA)

RSAs were done as described previously [26,29]. Schizonts were purified from an asynchronous parasite culture using a percoll gradient [133] and were allowed to invade fresh RBCs at 37°C for 3 hours after which they were synchronised with 5% sorbitol [134] to obtain 0–3 hour old rings. These rings were washed 3 times with medium and challenged with 700 nM DHA for 6 hours. Afterwards, the cells were washed three times in RPMI medium and the parasites were grown for another 66 hours. Finally, Giemsa smears were prepared in order to determine the parasite survival rate (parasitemia of viable parasites after DHA compared to parasitemia of non-DHA treated control). A survival rate of 1% parasite was considered the threshold for *in vitro* ART resistance [26]. Number of cells counted are indicated in the corresponding Figure legends. Addition of 250 nM rapalog or cytochalasin D (10μM final concentration) were added before or with the start of the RSA as indicated in the corresponding schemes and figure legends and lifted after the DHA pulse as described [29].

### Vesicle accumulation assay

The vesicle accumulation assay was adapted from [54]. Briefly, the number of vesicles per parasite were determined based on DIC images of synchronised trophozoites. For this, parasite cultures were two times synchronised (6–8 hours apart) using 5% sorbitol [134], split into two 2ml dishes of which one received rapalog to a final concentration of 250 nM, while the other culture served as control without rapalog and then grown for 16–24 hours to obtain trophozoite stages. Parasites were imaged in the DIC channel and the vesicles in the DIC images were counted. The assay was performed blinded and in at least three independent experiments (n of analysed cells indicated in the corresponding Figure legend).

### Bloated food vacuole assay / E64 hemoglobin uptake assay

The bloated food vacuole assay was performed as previously described [54]. Briefly, ring stage parasite cultures with an 8h time window were obtained using double 5% sorbitol synchronisation. Parasites were either split into two 2ml dishes of which one received rapalog to a final concentration of 250 nM, while the other culture served as control without rapalog for *Pf*KIC11 and *Pf*KIC12 (see experimental setup scheme in S4C and S5D Figs) and incubated at 37°C overnight, first incubated at 37°C overnight and split at the start of the assay for *Pf*MyoF (see experimental setup scheme in S3G Fig), or not split in case for bloated food vacuole assays with TGD cell lines. When parasites reached the young trophozoites stage, the medium was aspirated and 1 ml medium containing 33 μM E64 protease inhibitor (Sigma Aldrich) was added. The cells were cultured for 8 hours and then imaged. The DIC image was used for scoring bloated food vacuoles and determination of parasite and food vacuole size. For visualisation of bloated food vacuoles, the cells were stained with 4.5 mg/ml dihydroethidium (DHE) for 15 minutes at 37°C. The experiment was performed blinded and in at least three independent experiments.

### Invasion and egress assay

The invasion/egress assay was adapted from previously published assays [68,74]. Briefly, KIC11-2xFKBP-GFP-2xFKBP late schizonts were purified using Percoll followed by an incubation period of 30 minutes at 37°C in the shaking incubator (800 rpm) allowing the parasites to reinvade new erythrocytes. Subsequently, the cells were transferred to static standard culture conditions, incubated for 3.5 hours and sorbitol synchronization was performed resulting in tightly synchronized parasite culture with a time window of 4h. Parasites were divided into

two dishes and knock-sideways was induced by addition of rapalog to one dish. Both dishes were cultured at 37°C until Giemsa-stained smears were prepared at the time points 38–42 hpi ('pre-egress') and 46–50 hpi ('post-egress'). The numbers of ring and schizont stage parasites were determined by counting parasites in randomly selected fields of sight and used to calculate the percentage of ruptured schizonts and number of new rings per ruptured schizont in the 'post-egress' samples.

## Domain identification using AlphaFold

The predicted protein structures of all known K13-compartment members were downloaded from the AlphaFold Protein Structure Database (alphafold.ebi.ac.uk) [87], except for UBP1 which was not available. The structure for UBP1 was predicted in 6 parts, coving resides 1–640, 641–1280, 1281–1920, 1921–2560, 2561–2880 and 2881–3499 using ColabFold [142]. VAST searches (ncbi.nlm.nih.gov/Structure/VAST/vastsearch.html) [143] were performed on all structures. The top 3 hits for each protein and protein part were aligned with the search model using the PyMol command cealign (Schrödinger, USA). Similarities with RMSDs of under 4 Å over more than 60 amino acids are listed in the results. Domains that were not previously annotated in Interpro (as of April 2022) [95] or PlasmoDB v.57 [144] were considered as newly identified.

## Software

Statistical analyses were performed with GraphPad Prism version 8 (GraphPad Software, USA), microscopy images were processed in Corel Photo-Paint X6-X8 (https://www.coreldraw.com) or Fiji [145], plasmids and oligonucleotides were designed using ApE [146]. Protein structures were analysed and visualized using PyMol (Schrödinger, USA). Figures were arranged in CorelDraw X6-8.

## Supporting information

**S1 Fig. Validation of generated transgenic cell lines.** Confirmatory PCR of unmodified wild-type (WT) and transgenic knock-in (KI) / targeted-gene-disruption (TGD) cell lines to confirm correct genomic integration at the 3'- and 5'-end of the locus. Oligonucleotides used are listed in S2 Table. **(A)** MCA2[Y1344STOP]-GFP[endo]; **(B)** MCA2-3xHA[endo]; **(C)** MyoF-2xFKBP-GFP-2xFKBP[endo]; **(D)** MyoF-2xFKBP-GFP[endo]; **(E)** MyoF-3xHA[endo]; **(F)** KIC11-2xFKBP-GFP-2xFKBP[endo]; **(G)** KIC12-2xFKBP-GFP-2xFKBP[endo]; **(H)** PF3D7_0907200-2xFKBP-GFP-2xFKBP[endo]; **(I)** VPS51-2xFKBP-GFP-2xFKBP[endo]; **(J)** PF3D7_1365800-2xFKBP-GFP-2xFKBP[endo]; **(K)** UIS14-2xFKBP-GFP-2xFKBP[endo]; **(L)** PF3D7_1447800-2xFKBP-GFP-2xFKBP[endo]; **(M)** VPS51-TGD[endo]; **(N)** UIS14-TGD[endo]. Right panel in **(F and G)** Western Blot analysis of **(F)** MCA2[Y1344STOP]-GFP[endo] cell line using mouse anti-GFP to detect the tagged fusion protein (upper panel) and rabbit anti-aldolase to control for equal loading (lower panel)(expected molecular weight for MCA2[Y1344STOP]-GFP fusion proteins: 187 kDa) and **(G)** wildtype (3D7) and knock-in MCA2-3xHA[endo] cell line using mouse anti-HA to detect the tagged full-length protein (upper panel) and rabbit anti-aldolase to control for equal loading (lower panel) (expected molecular weight for MCA2-3xHA fusion protein: 281 kDa). Protein size is indicated in kDa.
(TIF)

**S2 Fig. Additional data for MyoF. (A)** Extended panel of live cell microscopy images of parasites expressing the MyoF-2xFKBP-GFP-2xFKBP fusion protein from the endogenous locus with episomally expressed mCherry-K13. Nuclei were stained with Hoechst. Scale bar, 5 μm.

**(B)** Localisation of MyoF-2xFKBP-GFP by live-cell microscopy across the intra-erythrocytic development cycle. Nuclei were stained with DAPI. Scale bar, 5 μm. **(C)** IFA microscopy images of acetone-fixed parasites expressing MyoF-3xHA with episomally expressed mCherry-K13 across the intra-erythrocytic development cycle. Nuclei were stained with DAPI. Scale bar, 5μm. Foci were categorized into 'overlap' (black), 'partial overlap' (dark grey), close foci (light blue) and 'non overlap' (light grey) in n = 31 parasites. Scale bar, 5μm. **(D)** Live-cell microscopy of knock sideways (+ rapalog) and control (without rapalog) MyoF-2xFKBP-GFP-2xFKBP$^{endo}$+1xNLSmislocaliser parasites at 0, 1, 2, 4 and 22 hours after the induction of knock-sideways by addition of rapalog. **(E)** Individual growth curves of MyoF-2xFKBP-GFP-2xFKBP$^{endo}$+1xNLSmislocaliser with (red) or without (blue) addition of rapalog shown in Fig 1F. Summary of individual growth curves shown as percentage of control parasites (without rapalog). Three (3D7, black) and eight (MyoF-2xFKBP-GFP-2xFKBP$^{endo}$, red) independent experiments. Error bars, mean ± SD. **(F)** Relative growth of synchronised MyoF-3xHA$^{endo}$ compared with 3D7 wild type parasites after two growth cycles. Each dot shows one of six independent growth experiments. P-values determined by one-sample t-test. **(G)** Number of vesicles per parasite in trophozoites determined by live-cell fluorescence microscopy (DIC) in 3D7 and MyoF-3xHA$^{endo}$ parasites. Three independent experiments with each time n = 27–38 (mean 31.7) parasites analysed per condition. Representative DIC images shown on the right. **(H)** Experimental setup of the bloated food vacuole assay shown in Fig 1H. **(I)** Bloated food vacuole assay with 3D7 parasites 8 hours after rapalog addition compared with controls (- rapalog). Cells were categorized as with 'bloated FV' or 'non-bloated FV' and percentage of cells with bloated FV is displayed; n = 2 independent experiments with each n = 26–40 (mean 31,3) parasites analysed per condition. Representative DIC and fluorescence microscopy images are shown. Parasite cytoplasm was visualized with DHE. **(J)** Representative images of bloated food vacuole assays with knock sideways (+ rapalog; bottom row) and control (without rapalog; top row) MyoF-2xFKBP-GFP-2xFKBP$^{endo}$+1xNLSmislocaliser parasites shown in Fig 1I and 1J. Scale bar, 5 μm. **(K)** Experimental setup of the RSA shown in Fig 1I. Schematic representation of the cell lines depicted above the corresponding panel. (TIF)

**S3 Fig. Additional data for KIC11. (A)** Live-cell microscopy of knock sideways (+ rapalog) and control (without rapalog) KIC11-2xFKBP-GFP-2xFKBP$^{endo}$ $^{ama1}$IMC1c-mSca_*nmd3*'-NLS-FRB-T2A-DHODH, KIC11-2xFKBP-GFP-2xFKBP$^{endo}$+ $^{ama1}$ARO-mSca_*nmd3*'-NLS-FRB-T2A-DHODH or KIC11-2xFKBP-GFP-2xFKBP$^{endo}$+ $^{ama1}$AMA1-mSca_*nmd3*'-NLS-FRB-T2A-DHODH parasites 40 hours after the induction of knock-sideways by addition of rapalog. Nuclei were stained with Hoechst; scale bar, 5 μm. **(B)** Quantification of KIC11 and K13 foci per cell in ring, trophozoite and schizont stage KIC11-2xFKBP-GFP-2xFKBP$^{endo}$ with episomally expressed mCherry-K13 parasites. N indicates the number of analysed KIC11-2xFKBP-GFP-2xFKBP parasites with episomally expressed mCherry-K13 per stage. **(C)** Extended panel of live cell microscopy images of parasites expressing KIC11-2xFKBP-GFP-2xFKBP$^{endo}$ with episomally expressed mCherry-K13 shown in Fig 2B. Nuclei were stained with DAPI. Scale bar, 5 μm except last column (zoom merozoite image): Scale bar = 1.25μm. Arrows are indicating categories from Fig 2C **(D)** Individual growth curves of knock sideways (+ rapalog) and control (without rapalog) of KIC11-2xFKBP-GFP-2xFKBP$^{endo}$+-1xNLSmislocaliser shown in Fig 2E. **(E)** Live-cell microscopy of knock sideways (+ rapalog) and control (without rapalog) KIC11-2xFKBP-GFP-2xFKBP$^{endo}$+1xNLSmislocaliser parasites 80 hours after the induction of knock-sideways by addition of rapalog. Scale bar, 5 μm. **(F)** Parasite stage distribution in Giemsa smears at the time points (average hours post invasion, h) indicated above each bar in tightly synchronised (±4h) KIC11-2xFKBP-GFP-

2xFKBP^endo+1xNLSmislocaliser parasites (rapalog addition at 4 hpi, 20 hpi, or 32 hpi and control) parasite cultures over two consecutive cycles (last time point in cycle 3). A second replicate is shown in Fig 2F. **(G)** Quantification of ruptured schizonts at 'post-egress' time point compared to 'pre-egress' time point in knock-sideways (+rapalog) compared to control (-rapalog) in KIC11-2xFKBP-GFP-2xFKBP^endo+1xNLSmislocaliser parasites from n = 3 independent experiments (indicated by different symbol shapes). P values were determined with ratio-paired t-test, data displayed as mean ±SD. Corresponding quantification of ring per ruptured schizont is shown in Fig 2H. **(H)** Experimental setup of the bloated food vacuole assay shown in Fig 2J. **(I)** Experimental setup of the RSA shown in Fig 2K. Schematic representation of the cell lines depicted above the corresponding panel.
(TIF)

**S4 Fig. Additional data for KIC12. (A)** Extended panel of KIC12-2xFKBP-GFP-2xFKBP^endo localisation by live-cell microscopy across the intra-erythrocytic development cycle. Nuclei were stained with Hoechst. Scale bar, 5 μm. Expanded panel of Fig 3B Foci were categorized into 'overlap' (black), 'partial overlap' (grey) and 'no overlap' (white) and shown as frequencies in the pie chart (n = 97 cells were scored from a total of five independent experiments). **(B)** Absence of KIC12-2xFKBP-GFP-2xFKBP^endo in free merozoites by live-cell microscopy. Nuclei were stained with Hoechst. Scale bar, 5 μm. **(C)** Individual growth curves of knock sideways (+ rapalog; red) and control (without rapalog; blue) KIC12-2xFKBP-GFP-2xFKBP^endo+-1xNLSmislocaliser shown in Fig 3D. **(D)** Individual growth curves of knock sideways (+ rapalog; red) and control (without rapalog; blue) KIC12-2xFKBP-GFP-2xFKBP^endo+-LYNmislocaliser shown in Fig 3D. **(E)** Experimental setup of the bloated food vacuole assay shown in Fig 3F. **(F)** Representative images of bloated food vacuole assay knock sideways (+ rapalog; bottom row) and control (without rapalog; top row) KIC12-2xFKBP-GFP-2xFKBP^endo+1xNLSmislocaliser Scale bar, 5 μm. **(G)** Representative images of knock sideways (+ rapalog, top row) and control (without rapalog; bottom row) KIC12-2xFKBP-GFP-2xFKBP^endo+LYNmislocaliser parasites from bloated food vacuole assay. Scale bar, 5 μm. **(H)** Experimental setup of the RSA shown in Fig 3L. Schematic representation of the cell lines depicted above the corresponding panel.
(TIF)

**S5 Fig. Additional data for MCA2. (A)** Localisation of MCA2-TGD-GFP by live-cell microscopy across the intra-erythrocytic development cycle. Nuclei were stained with DAPI. Scale bar, 5 μm. **(B)** IFA microscopy images of acetone-fixed parasites expressing MCA2-3xHA with episomally expressed mCherry-K13 across the intra-erythrocytic development cycle. Nuclei were stained with DAPI. Scale bar, 5μm. Foci were categorized into 'overlap' (black), 'partial overlap' (dark grey) and 'no overlap' (light grey) in n = 30 parasites. Schematic representation of the cell lines depicted above the corresponding panel. **(C)** Bloated food vacuole assay with 3D7, MCA2^Y1344STOP and MCA2-TGD parasites. Cells were categorized as with 'bloated FV' or 'non-bloated FV' and percentage of cells with bloated FV is displayed; n = 4 (3D7, MCA2^Y1344STOP) or n = 3 (MCA2-TGD) independent experiments with each n = 10–34 (mean 25.8) parasites analysed per condition. **(D)** Area of the FV, area of the parasite and area of FV divided by area of the corresponding parasites were determined. Mean of each independent experiment indicated by coloured symbols, individual data points by grey dots. Data presented according to SuperPlot guidelines [147]; Error bars represent mean ± SD. P-value determined by un-paired t-test. Area of FV of individual cells plotted versus the area of the corresponding parasite. Line represents linear regression with error indicated by dashed line. **(E)** Representative DIC images from S2C and S2D Fig are displayed. Schematic representation of the cell

lines depicted above the corresponding panel.
(TIF)

**S6 Fig. Candidate proteins not associated with the K13 compartment. (A)** Live cell microscopy images of parasites endogenously expressing VPS51-2xFKBP-GFP-2xFKBP with episomally expressed mCherry-K13 across the intra-erythrocytic development cycle. Scale bar, 5 μm. **(B)** Localisation of truncated VPS51TGD-GFP fusion protein by live-cell microscopy across the intra-erythrocytic development cycle. Nuclei were stained with DAPI. Scale bar, 5 μm. Schematic representation of the truncation strategy depicted above the panel, numbers indicating AA. **(C)** Relative growth of VPS51TGD and UIS14TGD parasites compared to 3D7 wild type parasites after two cycles. Four independent growth experiments. P-values determined by one-sample t-test. **(D)** Live cell microscopy images of parasites endogenously expressing UIS14-2xFKBP-GFP-2xFKBP with episomally expressed mCherry-K13 across the intra-erythrocytic development cycle. Scale bar, 5 μm. **(E)** Localisation of truncated UIS14TGD-GFP fusion protein by live-cell microscopy across the intra-erythrocytic development cycle. Nuclei were stained with DAPI. Scale bar, 5 μm. Schematic representation of the truncation strategy depicted above the panel, numbers indicating AA. **(F)** Live cell microscopy images of parasites endogenously expressing PF3D7_1447800-2xFKBP-GFP-2xFKBP with episomally expressed mCherry-K13 across the intra-erythrocytic development cycle. Nuclei were stained with DAPI. Scale bar, 5 μm. **(G)** Live cell microscopy images of parasites endogenously expressing PF3D7_0907200-2xFKBP-GFP-2xFKBP with episomally expressed mCherry-K13 across the intra-erythrocytic development cycle. Scale bar, 5 μm. Schematic representation of relevant features of each cell line depicted above the corresponding panel.
(TIF)

**S7 Fig. 3D7_1365800 an AH domain containing protein is dispensable for asexual parasite development. (A)** Live cell microscopy images of parasites endogenously expressing PF3D7_1365800-2xFKBP-GFP-2xFKBP by live-cell microscopy across the intra-erythrocytic development cycle. Nuclei were stained with DAPI. Scale bar, 5 μm. **(B)** Expression of PF3D7_1365800-2xFKBP-GFP-2xFKBP with episomally expressed mCherry-K13. Scale bar, 5 μm. **(C)** Live-cell microscopy of knock sideways (+ rapalog) and control (without rapalog) PF3D7_1365800-2xFKBP-GFP-2xFKBP[endo]+1xNLSmislocaliser parasites or PF3D7_1365800-2xFKBP-GFP-2xFKBP[endo]+LYNmislocaliser parasites 16 hours after the induction of knocksideways by addition of rapalog. Scale bar, 5 μm. **(D)** Individual growth curves of PF3D7_1365800-2xFKBP-GFP-2xFKBP[endo]+1xNLSmislocaliser or PF3D7_1365800-2xFKBP-GFP-2xFKBP[endo]+LYNmislocaliser parasites with (red) or without (blue) addition of rapalog. Relative growth of knock sideways (+ rapalog) compared to control (without rapalog) PF3D7_1365800-2xFKBP-GFP-2xFKBP[endo]+1xNLSmislocaliser (blue) or PF3D7_1365800-2xFKBP-GFP-2xFKBP[endo]+LYNmislocaliser (red) parasites over five days. Three (NLSmisloc) or six (LYNmisloc) independent growth experiments were performed and mean relative parasitemia ± SD is shown. **(E)** Live cell microscopy images of parasites expressing PF3D7_1365800-2xFKBP-GFP-2xFKBP with episomally expressed Golgi marker GRASP-mCherry. Schematic representation of the cell lines depicted above the corresponding panel. Scale bar, 5 μm.
(TIF)

**S8 Fig. Type of domain found in K13 compartment proteins coincide with functional group. (A)** Full domain names and Interpro domain numbers for each domain in Fig 5 with indication in which K13-compartment members these proteins occur. For each newly identified domain a brief summary of its reported function is given. Colours are as in Fig 5. **(B)** Full

length AlphaFold predictions for each K13-compartment member in which new domains were identified. New domains are coloured as in A and Fig 5. For UBP1 no prediction was available in the EMBL-AlphaFold database, and the structure was predicted in fragments as described in the methods. The predicted fragment containing the newly identified domain is shown. **(C)** Full length structure of PF3D7_1365800, with AH domain in green. PF3D7_1365800 AH domain in green aligned with the most similar domain from the PDB. PDB ID and alignment details are indicated beneath each set of aligned domains. Brief summary about the AH domain and its Interpro ID are given.
(TIF)

**S9 Fig. Targeted gene disruption of KIC4 and KIC5 has no effect in bloated food vacuole assays.** Bloated food vacuole assay with 3D7, KIC4-TGD and KIC5-TGD parasites. Cells were categorized as with 'bloated FV' or 'non-bloated FV' and percentage of cells with bloated FV is displayed; n = 3 independent experiments with each n = 20–38 (mean 26.7) parasites analysed per condition. Representative DIC are displayed. Area of the FV, area of the parasite and area of FV divided by area of the corresponding parasites were determined. Mean of each independent experiment indicated by coloured symbols, individual data points by grey dots. Data presented according to SuperPlot guidelines [147]; Error bars represent mean ± SD. P-value determined by paired t-test. Area of FV of individual cells plotted versus the area of the corresponding parasite. Line represents linear regression with error indicated by dashed line.
(TIF)

**S10 Fig. Knock sideways of KIC4. (A)** Live-cell microscopy of knock sideways (+ rapalog) and control (without rapalog) KIC4-2xFKBP-GFP-2xFKBP[endo]+ 1xNLS mislocaliser parasites 4 and 20 hours after the induction of knock-sideways by addition of rapalog. Scale bar, 5 μm. Graphs show relative growth of asynchronous KIC4-2xFKBP-GFP-2xFKBP[endo]+- 1xNLSmislocaliser plus rapalog compared with control parasites over five days (3 independent experiments). Mean relative parasitemia ± SD of plus rapalog vs control of the 3 experiments is shown in the fourth graph. **(B)** Bloated food vacuole assay with KIC4-2xFKBP-GFP-2xFKBP[endo]+1xNLSmislocaliser parasites 8 hours after inactivation of KIC4 (+rapalog). Cells were categorized as with 'bloated FV' or 'non-bloated FV' and percentage of cells with bloated FV is displayed; n = 3 independent experiments with each n = 19–30 (mean 21.4) parasites analysed per condition. Representative DIC are displayed. Area of the FV, area of the parasite and area of FV divided by area of the corresponding parasites were determined. Mean of each independent experiment indicated by coloured symbols, individual data points by grey dots. Data presented according to SuperPlot guidelines [147]; Error bars represent mean ± SD. P-value determined by paired t-test. Area of FV of individual cells plotted versus the area of the corresponding parasite. Line represents linear regression with error indicated by dashed line.
(TIF)

**S1 Table. List of putative K13 compartment proteins, with proteins selected for further characterization in this manuscript highlighted.** Ranked by enrichment: from high to low by the average log2 ratio normalized of four published Kelch13 BioID experiments [29].
(PDF)

**S2 Table. Oligonucleotides used for cloning and diagnostic genotyping PCR.**
(PDF)

**S1 Data. Excel tables containing the numerical values underlying the data in the figures.**
(XLSX)

## Acknowledgments

We thank Jacobus Pharmaceuticals for WR99210. DSM1 (MRA-1161) was obtained from MR4/BEI Resources, NIAID, NIH. We thank Tim Gilberger for providing the PhIL1mcherry plasmid.

## Author Contributions

**Conceptualization:** Tobias Spielmann.

**Formal analysis:** Sabine Schmidt, Jan Stephan Wichers-Misterek, Hannah Michaela Behrens.

**Funding acquisition:** Tobias Spielmann.

**Investigation:** Sabine Schmidt, Jan Stephan Wichers-Misterek, Hannah Michaela Behrens, Jakob Birnbaum, Isabelle G. Henshall, Jana Dröge, Ernst Jonscher, Sven Flemming.

**Project administration:** Tobias Spielmann.

**Resources:** Jakob Birnbaum, Ernst Jonscher, Sven Flemming, Paolo Mesén-Ramírez, Tobias Spielmann.

**Supervision:** Tobias Spielmann.

**Validation:** Carolina Castro-Peña.

**Visualization:** Sabine Schmidt, Jan Stephan Wichers-Misterek, Hannah Michaela Behrens, Isabelle G. Henshall, Tobias Spielmann.

**Writing – original draft:** Jan Stephan Wichers-Misterek, Hannah Michaela Behrens, Tobias Spielmann.

**Writing – review & editing:** Jan Stephan Wichers-Misterek, Hannah Michaela Behrens, Isabelle G. Henshall, Tobias Spielmann.

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
