## [Decision Letter · Decision Letter 0]

28 Aug 2023

Dear Dr. Spielmann,

Thank you very much for submitting your manuscript "The Kelch13 compartment is a hub of highly divergent vesicle trafficking proteins in malaria parasites" for consideration at PLOS Pathogens. As with all papers reviewed by the journal, your manuscript was reviewed by members of the editorial board and by several independent reviewers. In light of the reviews (below this email), we would like to invite the resubmission of a significantly-revised version that takes into account the reviewers' comments.

The revised version of the manuscript, which was originally reviewed through RevComm was reviewed by the original reviewers. As you can see in their comments two of the referees are satisfied with the significant amount of data included to address their concerns. Though, one reviewer still have major concerns primarily because they feel that some of the data regarding the characterized proteins does not support the endocytosis-ART resistance paradigm i.e. this reviewer argue that maybe not all proteins involved in endocytosis are linked with ART-resistance. I believe that these criticism should and could be addressed before making a final decision for the manuscript.

We cannot make any decision about publication until we have seen the revised manuscript and your response to the reviewers' comments. Your revised manuscript is also likely to be sent to reviewers for further evaluation.

Sincerely,

Ron Dzikowski

Academic Editor

PLOS Pathogens

James Collins III

Section Editor

PLOS Pathogens

Kasturi Haldar

Editor-in-Chief

PLOS Pathogens

orcid.org/0000-0001-5065-158X

Michael Malim

Editor-in-Chief

PLOS Pathogens

orcid.org/0000-0002-7699-2064

The revised version of the manuscript, which was originally reviewed through RevComm was reviewed by the original reviewers. As you can see in their comments two of the referees are satisfied with the significant amount of data included to address their concerns. Though, one reviewer still have major concerns primarily because they feel that some of the data regarding the characterized proteins does not support the endocytosis-ART resistance paradigm i.e. this reviewer argue that maybe not all proteins involved in endocytosis are linked with ART-resistance. I believe that these criticism should and could be addressed before making a final decision for the manuscript.

Reviewer's Responses to Questions

**Part I - Summary**

Reviewer #1: The revised version of the publication is much improved. Results are presented more clearly and structured. An enormous amount of data was generated and validated with solid experiments and the results will help the research community to further explore the endocytosis pathways in Plasmodium.

Reviewer #2: This paper investigates putative K13 interaction partners, identified in a previous study. Of the 10 investigated, 4 are at the K13 compartment, and two of which are involved in endocytosis. One (KIC11) is likely involved in invasion. 3 other proteins identified previously as being involved in ART resistance (MCA2, KIC4 and KIC5) do not appear to be involved in endocytosis. The paper is novel and significant with overall adequate execution. Some of their lines are suboptimal (TGDs and the MyoF work due to the poor growth of the parasites and possible compensatory effects), however the authors do their best to address this. The weaknesses lie in some of the data interpretation and structural domain assignment, which suggests a lot of the K13 compartment proteins contain endocytic domains - however this is speculative and then the two candidates that they look at end up not being involved in endocytosis. I find it peculiar that some of the endocytosis data is not believed by the authors due to it not being supportive of their domain assignment or previous models.

Reviewer #3: The manuscript by Schmidt et al. follows up on work that identified the molecular mechanism of ART resistance in P. falciparum (PMID: 31896710), by further investigating ten genes of the identified K13/Eps15-related “proxiome”. The authors link MCA2 to ART resistance in vitro, while the proteins MyoF and KIC12 are involved in endocytosis but do not confer in vitro ART resistance. Characterization of KIC11, which partially colocalizes with K13 in trophozoites/schizonts, indicates an important function in IDC unrelated to endocytosis. Five analyzed genes however do not colocalize with the K13 compartment, while a sixth was refractory to endogenous tagging. Using AlphaFold prediction the authors identify protein domains in K13 compartment constituents, which have not been recognized before due to their unusual arrangement and low level of primary sequence conservation.

Endocytosis is insufficiently understood in Plasmodium and this manuscript makes an important contribution by further dissecting the unusual protein machinery employed by the parasite. Overall this study is of high quality and the presented experiments are well controlled for. The authors adequately addressed my previous comments and I only have some small text edit suggestions.

**Part II – Major Issues: Key Experiments Required for Acceptance**

Reviewer #1: No major issues in the revised publication.

Reviewer #2: First of all I would like to thank the authors for all the extra work that they have done for the manuscript. It has improved the overall quality and accessibility of the data. However, I am concerned by the authors' resistance in believing the endocytosis assay data for MCA2, KIC4 and KIC5 TGD lines. It appears that the correct controls have been included and that there are no endocytosis defects. It could be argued that the data on KIC5 is questionable because of the major growth defect of this TGD line and the inability to knock the protein sideways, but MCA2 and KIC4 do not have growth defects, so how can the authors still not support this data? I also find it particularly concerning that despite the fact that they generated a convincing KIC4 KS line , they still chose not to believe the fact that KIC4 is not involved in endocytosis and chose not to put the data in the paper all together. Is this because it weakens their structural domain prediction data or is it because the MCA2 and KIC4 data does not support the model put forward by Birnbaum et al., 2020 paper that links the endoytosis/ART resistance link? Moreover if the authors don't believe some of their data because their TGD lines have growth defects, why do they believe the MyoF data? This is a major contradiction and extremely concerning and could be evidence of cherry picking data. I strongly urge the authors to include the KIC4 KS data in the manuscript and highlight how an ART resistance protein might not be involved in endocytosis (as is the case for MCA2 truncated line and KIC4) where both show ART resistance and no involvement in endocytosis.

I would consider re-naming the paper - only 2 of the 12 candidates you looked at are involved in endocytosis or the K13 compartment. Maybe consider something along the lines of divergent functions of K13 compartment proteins including proliferation and endocytosis.

Figure 1C: the merozoite image is not representative of true merozoites, rather a schizont that is beginning to rupture. Please amend of the label of this subpanel or change the image to individual merozoites.

Line 250: the authors state here 'speaking against other myosins taking over the MyoF endocytosis function in rings'. This suggests MyoF is involved in endocytosing in rings - when the authors have not shown that MyoF is acting in rings, how can the authors state this? Especially considering MyoF is not involved in ART resistance how can the authors make these claims?

Many of the figures, namely figures 1 & 2 are very difficult to follow since the figures are not in order and jump up and down the page in terms of order. Please revise this so that it is easier for the reader to follow the order of the figures.

Reviewer #3: none

**Part III – Minor Issues: Editorial and Data Presentation Modifications**

Reviewer #1: No minor issues in the revised publication.

Reviewer #2: Line 59: typo - Papua New Guinea not Papa New Guinea

Reviewer #3: Edits:

-line 59: Change Papa New Guinea to Papua New Guinea

-line 149: Change location to localization

-line 350: Change location to localization

-line 435-436: “….verified by Western blot…” should be “verified by western blot…”

-line 1056: Means of independent experiments are presented as colored dots in Fig 1J and not as triangles as stated in the legend, please adjust either text or figure.

-line 1158: Same as above, please adjust for Fig 3G/I

-line 1185: Please fix typo in metacaspase

PLOS authors have the option to publish the peer review history of their article (what does this mean?). If published, this will include your full peer review and any attached files.

Reviewer #1: No

Reviewer #2: No

Reviewer #3: No
---

## [Decision Letter · Decision Letter 1]

9 Nov 2023

Dear Dr. Spielmann,

We are pleased to inform you that your manuscript 'The Kelch13 compartment contains highly divergent vesicle trafficking proteins in malaria parasites' has been provisionally accepted for publication in PLOS Pathogens.

Best regards,

Ron Dzikowski

Academic Editor

PLOS Pathogens

James Collins III

Section Editor

PLOS Pathogens

Kasturi Haldar

Editor-in-Chief

PLOS Pathogens

orcid.org/0000-0001-5065-158X

Michael Malim

Editor-in-Chief

PLOS Pathogens

orcid.org/0000-0002-7699-2064

Reviewer Comments (if any, and for reference):

Reviewer's Responses to Questions

**Part I - Summary**

Reviewer #2: (No Response)

**Part II – Major Issues: Key Experiments Required for Acceptance**

Reviewer #2: (No Response)

**Part III – Minor Issues: Editorial and Data Presentation Modifications**

Reviewer #2: (No Response)

PLOS authors have the option to publish the peer review history of their article (what does this mean?). If published, this will include your full peer review and any attached files.

Reviewer #2: No

---

## [Editor Report · Acceptance letter]

27 Nov 2023

Dear Dr. Spielmann,

We are delighted to inform you that your manuscript, "The Kelch13 compartment contains highly divergent vesicle trafficking proteins in malaria parasites," has been formally accepted for publication in PLOS Pathogens.

Best regards,

Kasturi Haldar

Editor-in-Chief

PLOS Pathogens

orcid.org/0000-0001-5065-158X

Michael Malim

Editor-in-Chief

PLOS Pathogens

orcid.org/0000-0002-7699-2064